# Low rates of mutation in clinical grade human pluripotent stem cells under different culture conditions

Oliver Thompson[1], Ferdinand von Meyenn [2,3,6], Zoe Hewitt [1], John Alexander [1,7], Andrew Wood [1], Richard Weightman [1], Sian Gregory [1], Felix Krueger [4], Simon Andrews[4], Ivana Barbaric[1], Paul J. Gokhale [1], Harry D. Moore[1], Wolf Reik[2,5], Marta Milo [1], Serena Nik-Zainal [5,8,9], Kosuke Yusa [5,10 ✉] & Peter W. Andrews [1 ✉]

The occurrence of repetitive genomic changes that provide a selective growth advantage in pluripotent stem cells is of concern for their clinical application. However, the effect of different culture conditions on the underlying mutation rate is unknown. Here we show that the mutation rate in two human embryonic stem cell lines derived and banked for clinical application is low and not substantially affected by culture with Rho Kinase inhibitor, commonly used in their routine maintenance. However, the mutation rate is reduced by >50% in cells cultured under 5% oxygen, when we also found alterations in imprint methylation and reversible DNA hypomethylation. Mutations are evenly distributed across the chromosomes, except for a slight increase on the X-chromosome, and an elevation in intergenic regions suggesting that chromatin structure may affect mutation rate. Overall the results suggest that pluripotent stem cells are not subject to unusually high rates of genetic or epigenetic alterations.

[1] The Centre for Stem Cell Biology, Department of Biomedical Science, University of Sheffield, Western Bank, Sheffield S10 2TN, UK. [2] Epigenetics Programme, Babraham Institute, Cambridge CB22 3AT, UK. [3] Department of Medical & Molecular Genetics, King's College London, London SE1 9RT, UK. [4] Bioinformatics Group, Babraham Institute, Cambridge CB22 3AT, UK. [5] Wellcome Sanger Institute, Wellcome Genome Campus, Hinxton, Cambridge CB10 1SA, UK. [6] Present address: Institute of Food, Nutrition and Health, ETH Zurich, 8603 Schwerzenbach, Switzerland. [7] Present address: Breast Cancer Now Toby Robins Research Centre, The Institute of Cancer Research, London, UK. [8] Present address: Academic Laboratory of Medical Genetics, Cambridge University Hospitals NHS Foundation Trust, Box 238, Lv6 Addenbrooke' Treatment Centre, Cambridge Biomedical Research Campus, Cambridge CB2 0QQ, UK. [9] Present address: MRC Cancer Unit, University of Cambridge, Hutchinson/MRC Research Centre, Box 1297, Cambridge Biomedical Campus, Cambridge CB2 0XZ, UK. [10] Present address: Institute for Frontier Life and Medical Sciences, Kyoto University, Kyoto 606-8507, Japan. ✉email: k.yusa@infront.kyoto-u.ac.jp; p.w.andrews@sheffield.ac.uk

The presence of mutations in human pluripotent stem cells (PSC), whether embryonic stem (ES) cells or induced pluripotent stem (iPS) cells, is a concern for their safe use in therapeutic applications. Indeed, in one case, a potential trial of retinal pigment cells from an autologous iPS cell line was abandoned because the cells carried a mutation of unknown significance[1]. Certainly some such variants are likely to have been present in the embryos or somatic cells from which particular PSC were derived and can be classed as 'variants of origin'[2,3]. However, the propensity of PSC to acquire genetic variants on prolonged passage poses additional concerns, not only because of the difficulty for their early detection[4], but also their selective growth advantages might presage malignant potential[5].

Acquired genetic variants of human ES cells were first noticed as non-random gains of particular chromosomes or fragments of chromosomes detected by G-banding karyotypes[6,7]. Over succeeding years these observations were repeated in many laboratories as it became clear that certain chromosomal regions of human PSC, whether ES or iPS cells, were particularly subject to gains, notably chromosomes 1q, 12p, 17q and 20q, and the X chromosome, or losses, notably 10p, 18q and 22p[2]. At the same time, other chromosomes, notably chromosome 4, appeared impervious to any gains or losses. More refined analyses, for example using SNV arrays[2,8], also began to reveal small structural variants not readily detectable by G-banding cytogenetics. For example, a small variable length CNV was found in the proximal region of chromosome 20q in 20 human ES cell lines that were otherwise scored as normal, diploid cells by G-banding[2,9,10]. More recently, single base-pair mutations have been reported in the *TP53* gene of several human ES cell lines[11,12].

It seems likely that the repetitive, non-random nature of many, if not all, acquired mutations observed in human PSC results from their conferring a selective growth advantage. Certainly, chromosomal variants when initially observed in a small proportion of cells in a culture commonly come to predominate within very few passages, while experiments in which small numbers of variant cells have been mixed with their normal counterparts confirm the strong selective growth advantage of the variants[13]. Time lapse imaging of the growth patterns of variant and normal cells also indicates marked effects on the ability of the cells to form viable long-term colonies after passaging by overcoming multiple bottlenecks that restrict the ability of normal cells to proliferate[14]. Further, in the minimal amplicon of the chromosome 20 CNV, it has been possible to identify the likely driver gene, *BCL2L1*, overexpression of which inhibits apoptosis and increases survival of the variant cells[15,16]. Likewise, the mutations in *TP53* are likely to provide a growth advantage by suppressing apoptosis[11,12].

There have been many estimates of mutation rate in the germline and soma, although finding consensus in the reported rates is confounded by the variety of experimental and analytical methodologies used in their calculation. One recent study cites rates of $3.3 \times 10^{-11}$ and $2.66 \times 10^{-9}$ mutations per base-pair, per mitosis, in the germline and soma, respectively[17]. By comparison, Rouhani et al 2016[3] estimated the mutation rate in two human iPS cell lines and one widely used human ES cell line (H9), as $0.18 \times 10^{-9}$ mutations per base-pair, per cell division, whereas the corresponding mutation rate in somatic cells was ten-fold higher. In another study of one human iPS cell line[18] estimated a rate of $3.5 \pm 0.5$ base-pair substitutions per population doubling—equivalent to about $1 \times 10^{-9}$ mutations per base-pair, per cell division. Still, little detail is known of the mutation rates in PSC, which might arise from erroneous repair, or from defects in mitosis, for example, leading to chromosome non-dysjunction. Further, the possibility that some repetitive genomic variants reflect hotspots for chromosome rearrangements or other mutations cannot be excluded. PSC are one of the few 'normal' diploid cell types that do not undergo senescence and can be maintained indefinitely in vitro. Other diploid somatic cells undergo senescence, whereas other easily accessible cells that can be grown indefinitely are likely to be transformed cancer cells. Further, cell cycle control in PSC differs with respect to the lack of key checkpoints, notably the G1/S checkpoint[19], or the CHK1 checkpoint in S-Phase DNA replication resulting in apoptosis of PSC in response to DNA replication stress, in contrast to somatic cells[20]. This might reflect the relation of PSC to the rapidly dividing pluripotent cells of the early embryo for which there may be a survival advantage if cells suffering DNA damage undergo apoptosis rather than repair the damage.

Most studies in human PSC to date have been concerned with mutations providing a selective advantage, as these are the most frequently and easily detected when screening cell lines. However, estimating the underlying mutation rate is more difficult and can be confounded by the emergence of an advantageous or cell-lethal mutation, both of which could bias the mutational load of the population. To de-couple mutation from selection it is necessary to distinguish acquired de novo mutations, occurring during prolonged cell culture, from variants-of-origin mutations (i.e. those already present in the parental cells) and to estimate the frequency of their appearance before the selective overgrowth of any advantageous mutations.

In this study, to determine the rate and types of mutation that occur during culture of human PSC, we have used a cloning strategy coupled with whole-genome-, whole-genome bisulfite- and RNA-sequencing to compare two well characterised, clinical grade human ES cell lines, as well as the effects of culture in the presence of a Rho-kinase inhibitor, now commonly used in routine culture of human PSC, and of culture under low oxygen (5% $O_2$) conditions. We have tested whether the mutations are randomly distributed throughout the genome or clustered in regions related to chromatin structure and gene expression. Our results indicate that the mutation rate is low compared to estimates in somatic or cancer cells and can be further reduced by culture under low oxygen conditions. Furthermore, the mutational signature of human PSC is similar to that of other cultured human cells, but low oxygen culture does alter the frequency of some types of base-pair change associated with oxidative damage.

## Results

**Experimental design**. For these experiments, we chose two male human ES cell lines, MShef4 and MShef11 that we had derived under GMP-like conditions suitable for their potential clinical use in regenerative medicine. During their initial derivation, banking and characterisation, no karyotypic variants of MShef4 had been observed, and it was designated, a priori, as a genetically more stable line. By contrast, several cultures of MShef11, though not those used to produce master and working banks, were found to contain chromosomal aberrations (unpublished observations) and so this line was designated, a priori, as genetically less stable. To assess the mutation rates in these cells, the strategy we adopted was to produce a clonal subline that was karyotypically diploid, grow that clone for a defined period, and then produce a series of subclones that were subjected to whole-genome sequencing, as well as whole-genome bisulfite sequencing and RNA sequencing (Fig. 1).

**Mutation rate between cell lines and growth-conditions**. Whole-genome sequence (WGS) data were analysed using the CaVEMan variant detection algorithm (Cancer Genome Project, https://github.com/cancerit/CaVEMan). Following quality control

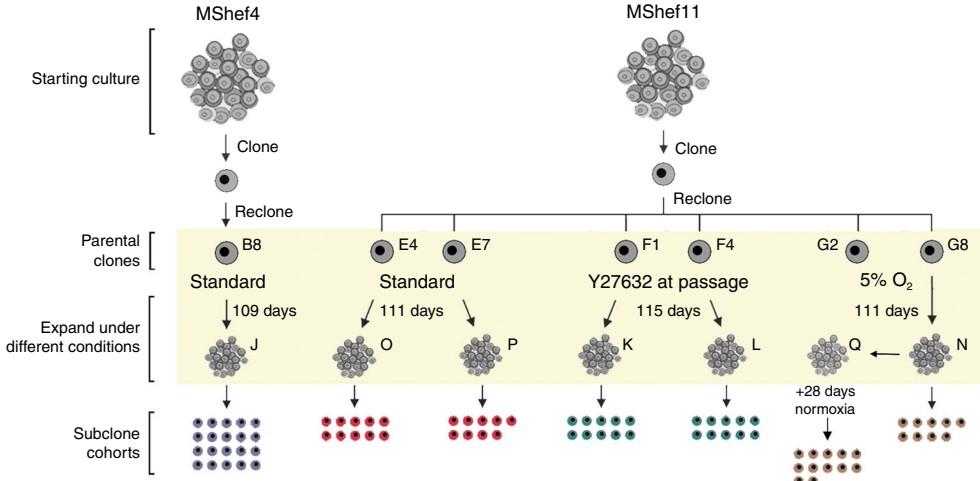

**Fig. 1 Schematic representation of the experimental design.** Schematic representation of the experimental regime, in which MShef4 and MShef11 human ES cell lines were cultured for an extended period under different growth-conditions. Stock cultures of MShef4 and MShef11 were cloned by single-cell deposition to ensure a homogeneous starting point. A clone of each was then recloned to generate Parent Clones that were then cultured under the different growth-conditions for a period exceeding three months—MShef4 clone B8, standard conditions, 109 days (19 passages); MShef11 clones E4 & E7, standard conditions, 111 days (25 passages); MShef11 clones F1 & F4, standard conditions + Y27632, 115 days, (25 passages); MShef11 clone G8 low oxygen, 111 days (25 passages) to give subclone cohort N, and 111 days + 28 days in standard oxygen (25 + 6 passages) to give subclone cohort Q. At the end of the expansion period cultures were recloned again into standard conditions to obtain subclone cohorts (J, O, P, K, L, N and Q, respectively, as shown), from which genetic material was extracted as early as possible (5 passages) for sequencing analysis.

12,555 SNVs were identified for subsequent analysis (see Methods of details).

The rate of somatic substitution mutation in each cohort of MShef4 and MShef11 subclones was assessed by calculating the number of de novo SNVs acquired per day of culture, per base-pair of sequenced haploid genome[21] (Supplementary Data 1). The mutations rates for MShef4 and MShef11 grown in standard conditions were both low ($0.37 \times 10^{-9}$ and $0.28 \times 10^{-9}$ SNVs per day, respectively) and not significantly different ($P = 0.084$) (Fig. 2a). The mutation rate for MShef11 in the presence of Y27632 was not significantly different, at $0.3 \times 10^{-9}$ SNVs per day, per base-pair ($P = 0.6$). However, MShef11 grown under low oxygen did have a significantly lower mutation rate compared to standard conditions ($P = 0.000034$) with a median of $0.13 \times 10^{-9}$ SNVs per day per base-pair, a reduction of 54%.

**Distribution of base-pair mutations across the genome.** To determine whether mutations occurred randomly and were distributed uniformly across the genome, or if there was an enrichment of mutations on particular chromosomes, we quantified the haploid mutation rate per chromosome, per day, per sequenced base-pair[21] and compared each to the genome-wide mutation rate. Due to a high level of noise generated from aligning and mapping mutations to the Y chromosome, we excluded this from this analysis. When we applied paired Mann-Whitney testing with Bonferroni correction to compare the median mutation rates of each chromosome to the respective genome-wide rate under each growth condition, none of the autosomes showed significant deviation from the genome-wide rate. However, the X chromosome showed an elevated mutation rate in all growth conditions (Fig. 2b), although this was significant only in the MShef4 standard and MShef11 low oxygen conditions ($P = 0.007$; $P = 0.034$). Nevertheless, when data from all growth conditions were combined into a single dataset, the mutation rate of genes on the X chromosome, but on none of the autosomes, continued to show a small but significantly ($p < 0.05$) elevated mutation rate.

We next measured the haploid mutation rates of intergenic, intronic and exonic regions, as annotated in the GRCh37 genome assembly, and normalised to the base-pair content of each class of feature (Fig. 2c). For these three features there was no difference in mutation rate between MShef11 grown in standard conditions or with Y27632, but the mutation rates of all features were reduced in MShef11 grown under low oxygen (intergenic $P = < 0.000049$; intronic $P = 0.00006$; exonic $P = 0.043$). There was a significant difference between the intronic mutation rate of MShef4 and MShef11 grown in standard conditions, with MShef4 having the higher mutation rate ($P = 0.04$). Within each growth-condition group, the median mutation rates of introns and exons were similar, but intergenic DNA had approximately double the mutation rate of exons or introns. This may be a result of preferential DNA repair in genic regions, as has been reported by others[22,23]. We asked if the elevated mutation rates seen on the X chromosome was due to a high proportion of intergenic DNA, but it does not show an unusually high intergenic DNA content (Supplementary Fig. 5a, upper panel) suggesting that the elevated mutation rate of this chromosome is not linked to its intergenic content alone. We also considered the GC nucleotide content of each chromosome, given the higher mutability of GC compared to AT nucleotides. Again, the X chromosome does not possess an unusually high GC content (Supplementary Fig. 5a, lower panel).

We also measured the mutation rates within a variety of other genomic features (Supplementary Fig. 5b). In most cases, we found a similar pattern of difference between groups, with subclones derived from low oxygen parental clones displaying a significantly lower mutation rate than other groups. Two exceptions that showed slightly elevated mutation rates across all groups were CpG islands and transcription factor-binding sites (TFBS). Both showed a slightly higher mutation rate than other features (with the exception of intergenic regions), with no significant differences in mutation rates between groups, suggesting that such genomic regions are particularly susceptible to mutation. However, given the small number of subclones that acquired mutations in these regions we could not draw further conclusions on the significance of this finding. The mutation rates

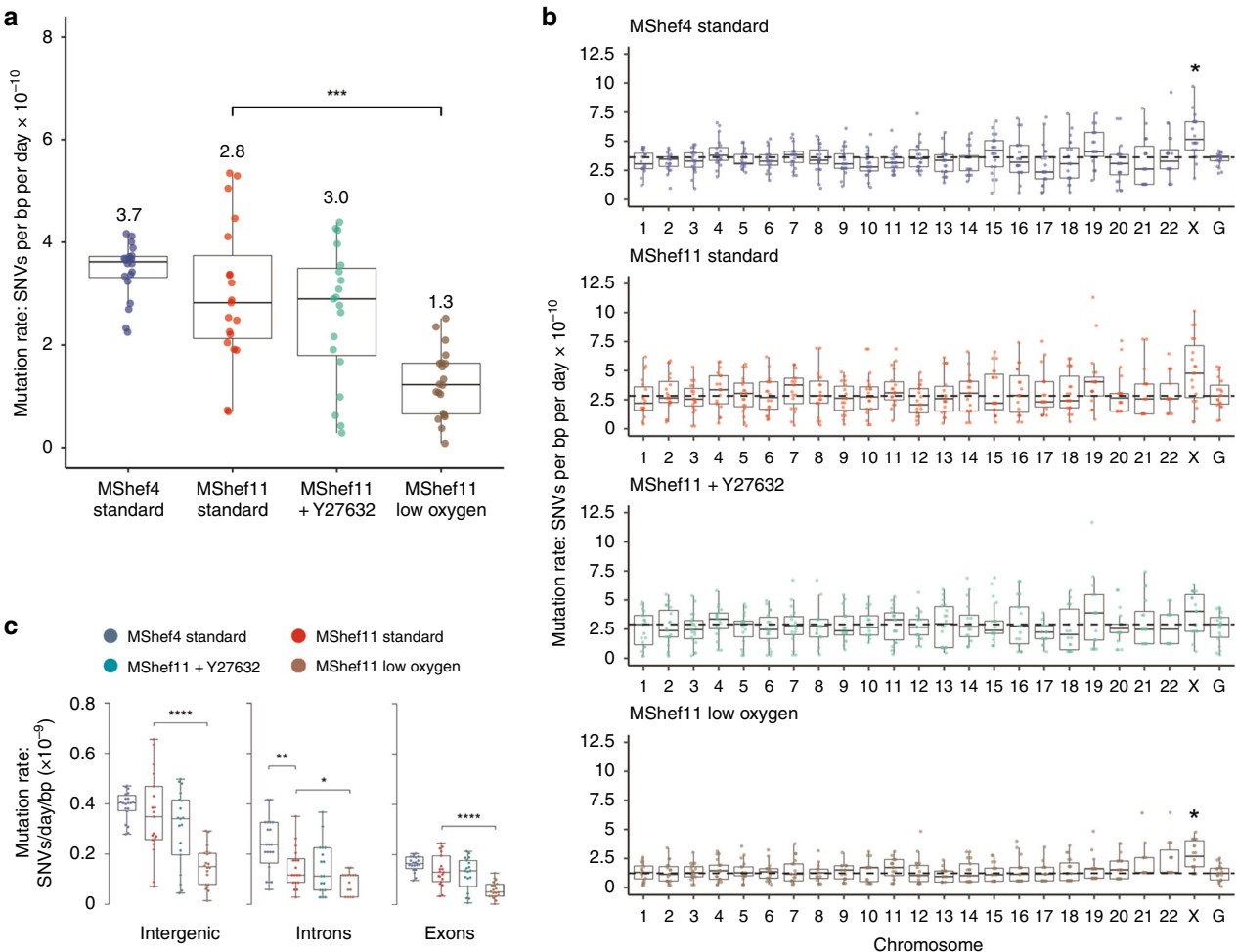

**Fig. 2 Mutation rate and genomic distribution of single-nucleotide variant mutations in different growth-conditions.** Box and whisker plots showing haploid mutation rates and genomic distribution of single-nucleotide variant mutations in subclones of MShef4 ($N = 20$) and MShef11 ($N = 19$) grown under standard conditions, and subclones of MShef11 cultured with the rho-kinase inhibitor Y27632 at passage ($N = 20$) or cultured in low oxygen ($N = 21$). Boxes represent 25th–75th percentiles; whiskers show the min and max ranges; horizontal lines indicate the median values. **a** The median mutation rates (shown below each plot) for MShef4 and MShef11, cultured in standard conditions, and MShef11 cultured with the rho-kinase inhibitor Y27632 were not significantly different. However, MShef11 cultured in low oxygen exhibited a ~54% lower mutation rate (unpaired two-tailed $t$-tests with Welch's correction; $P = < 0.0001$. ns: $P > 0.05$; $*P \leq 0.05$; $**P \leq 0.01$; $***P \leq 0.001$; $****P \leq 0.0001$. **b** Across all chromosomes only the X-chromosome mutation rate deviated from the genome-wide rates, being elevated in all growth conditions, although only significant in MShef4 and MShef11 under low oxygen, assessed by independent pairwise two-tailed Mann-Whitney tests comparing the median genome-wide rate in each condition with that of each chromosome (Bonferroni-adjusted $P = < 0.007$; $P = < 0.034$). When data from all growth conditions were combined, the elevated X-chromosome mutation rate remained significant ($P = < 0.05$). ns: $P > 0.05$; $*P \leq 0.05$; $**P \leq 0.01$; $***P \leq 0.001$; $****P \leq 0.0001$. **c** Across coding and non-coding regions, normalised to the genomic DNA content of each class of region, under each growth-condition intergenic DNA showed higher mutation rates than exons and introns. For intergenic DNA: MShef11 standard vs MShef11 low oxygen $P < 0.001$; For intronic DNA: MShef4 standard vs. MShef11 standard $P = 0.004$, MShef11 standard vs MShef11 low oxygen $P = 0.017$; For exonic DNA: MShef11 standard vs MShef11 low oxygen $P < 0.0001$. Asterisks indicate level of significance between groups assessed by independent pairwise two-tailed Mann-Whitney testing. ns: $P > 0.05$; $*P \leq 0.05$; $**P \leq 0.01$; $***P \leq 0.001$; $****P \leq 0.0001$. Source data are provided in Supplementary Data 1.

over a variety of histone marks were also calculated (Supplementary Fig. 5b). In the case of three types of histone mark (H3K4me1, H3K4me3, and H3K36me3), MShef4 had slightly higher though significant mutation rate than MShef11 grown under standard conditions ($P = 0.029$; $P = 0.008$; $P = 0.023$). In two cases (H3K9me3 and H3K27me3) MShef11 grown in low oxygen had a significantly lower mutation rate than MShef11 grown in standard conditions ($P = 0.0008$; $P = 0.013$).

**Pattern of mutational pressure and mutation signatures.** It is possible to extract unique mutational signatures from WGS data by examining the frequencies of different types of base substitution and the up- and downstream context surrounding the

SNV[24,25] (Supplementary Data 2). These signatures can provide information about the type of mutational stress experienced by cells.

Low-resolution mutation spectra showed that MShef4 and MShef11 grown in standard conditions or with Y27632 had a very similar pattern of base substitution. However, subclones derived from low oxygen culture had a slightly different pattern of substitution mutations compared to the other conditions (Fig. 3a) with significantly fewer C > A transversions ($P = 0.047$) and significantly more T > C transitions ($P = 0.045$) compared to standard conditions

High-resolution spectra (considering base-pairs flanking each mutation) produced a mutation profile that was similar between

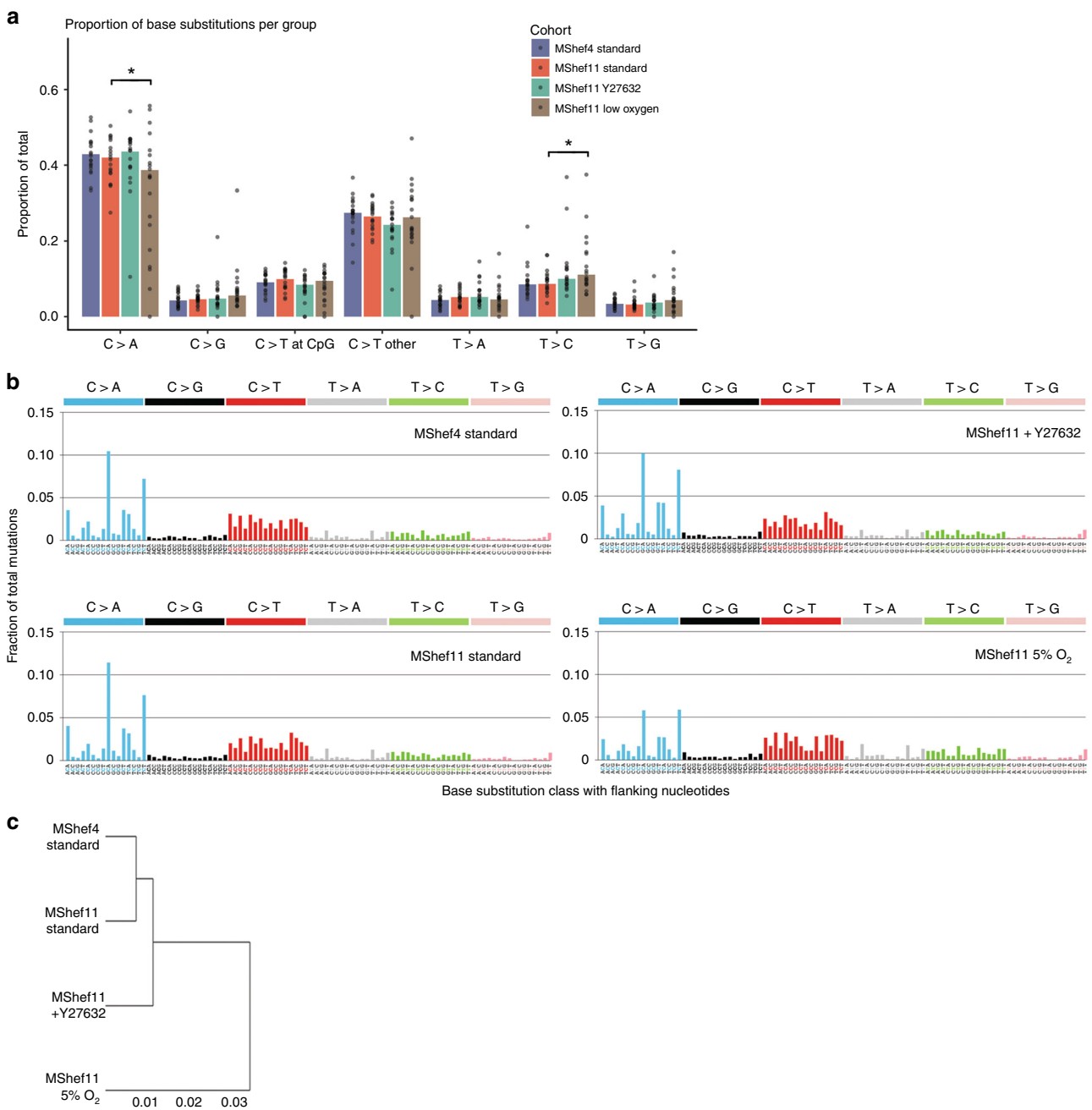

**Fig. 3 Pattern of base substitutions and mutation signatures. a** Bar and dot chart showing low-resolution mutational spectra and proportions of each of seven point mutation classes detected in MShef4 standard ($N = 20$), MShef11 standard ($N = 19$), MShef11 + Y27632 ($N = 20$), and MShef11 low oxygen ($N = 21$) subclones. Bars indicate the mean proportion of each class of mutation for each cell line/growth-condition group; dots indicate individual subclones in each group. C > A transversions and C > T transitions are most prevalent in the data. A two-tailed *t*-test showed MShef11 cultured in low oxygen to have a significant reduction in C > A transversions compared to standard conditions ($P = 0.037$) whilst Wilcoxon testing also showed a small increase in T > C transitions ($P = 0.045$). **b** High-resolution mutational spectra derived from the combined mutation data all subclones grown in each growth-condition. Each of the six possible point mutations is subdivided into 16 classes on the basis of the 5' and 3' nucleotides flanking the mutation, resulting in 96 possible substitution classes. C:G > A:T and C:G > T:A mutations are most prevalent in the data. These spectra can be correlated with 30 mutation signatures annotated in the Catalogue of Somatic Substitutions in Cancer (COSMIC) database[53] to explore the aetiology of mutation. **c** Dendrogram showing the similarity of all growth-condition groups based on their mutational profiles. MShef4 and MShef11 cultured in standard conditions exhibited the most similar mutational profiles, whereas the low oxygen condition is the most dissimilar in mutation profile compared to all other groups. Source data are provided in Supplementary Data 2.

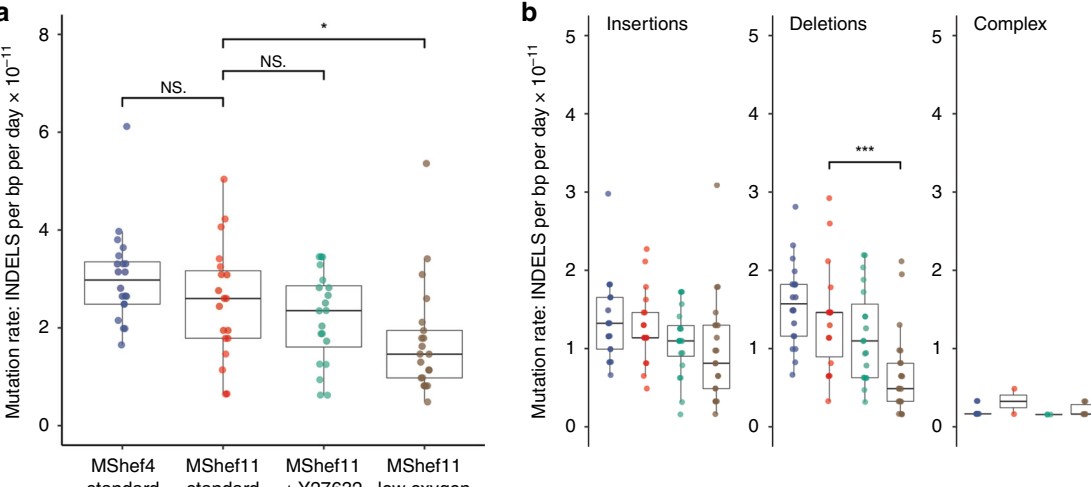

**Fig. 4 Mutation rate and genomic distribution of insertion and deletion mutations in different growth-conditions. a** Box and whisker plots showing haploid mutation rates all INDELs in subclones derived from clones grown in different conditions. MShef11 subclones derived from low oxygen ($N = 21$) showed a significantly reduced INDEL mutation rate compared to subclones from standard conditions ($N = 19$) (assessed by unpaired two-tailed $t$-test with Welch's correction; $P = < 0.02$). Taken together, the mutation rates of INDELs were approximately 10-fold lower than the mutation rates of single-nucleotide variants. Boxes represent the 25th–75th percentiles of the data; whiskers show the min and max range of the data; horizontal lines indicate the median value. ns: $P > 0.05$; *$P \leq 0.05$; **$P \leq 0.01$; ***$P \leq 0.001$; ****$P \leq 0.0001$. **b** Box and whisker plots showing haploid mutation rates of insertion, deletion and complex (mixed insertion and deletion) mutations in cells grown in different conditions. MShef11 subclones derived from low oxygen ($N = 21$) showed a significantly reduced number of deletions compared to those from standard conditions ($N = 19$) (assessed by unpaired two-tailed $t$-tests with Welch's correction; $P = < 0.0008$). The mutation rates of complex INDELs (mixed insertion and deletion) were 10-20X lower than the rates of insertions or deletions across all cell lines and growth conditions. Boxes represent the 25th–75th percentiles of the data; whiskers show the min and max range of the data; horizontal lines indicate the median value. ns: $P > 0.05$; *$P \leq 0.05$; **$P \leq 0.01$; ***$P \leq 0.001$; ****$P \leq 0.0001$. Source data are provided in Supplementary Data 3.

cell lines and growth conditions, with a predominance of C > A transversions (particularly in the *GCA* and *TCT* context) and C > T transitions (Fig. 3b). When we compared the mutational profile of each condition with 30 annotated in the Catalogue of Somatic Mutations in Cancer (COSMIC) database, our data correlated most closely with COSMIC signature 18 (cosine correlation = 0.873), which is associated with oxidative stress and is a hallmark of in vitro cell culture[25,26]. The reduced C > A component observed in MShef11 under low oxygen fits with a model of reduced oxidative stress, as oxidative species predominantly affect guanine (which is captured in the C > A class). Similarity-clustering showed that MShef4 and MShef11 grown in standard conditions acquired the most similar mutation profiles, followed by MShef11 cultured with Y27632, and the low oxygen condition as an outgroup (Fig. 3c) indicating that low oxygen culture induced the largest difference in mutational profile of all conditions tested.

**INDELs and structural variants**. To detect INDELs we used the *PINDEL* INDEL-calling algorithm (github.com/genome/pindel). Following quality control and excluding calls with length greater than 100 bp, as well as 291 calls that were recurrent in subclones of the same cohort, a total of 1171 de novo INDELs remained for analysis including single-base INDELs [572 insertions; 573 deletions; 26 complex] (Supplementary Figures 1–4; Supplementary Data 3). Out of 1171 INDELs, 578 are single-base INDELs of which 350 are insertions and 228 are deletions. Taken together, the median INDEL mutation rate was ~10-fold lower than that of base-pair substitutions (Fig. 4a). As with SNVs, we observed a lower mutation rate in MShef11 grown in low oxygen compared to MShef11 grown under standard conditions ($0.15 \times 10^{-10}$ vs $0.26 \times 10^{-10}$ INDELs per day, per base-pair; $P = 0.02$) and we observed a significant difference in median deletion mutation rates between MShef11 grown under low oxygen vs. MShef11

grown in standard conditions ($0.05 \times 10^{-10}$ vs $0.16 \times 10^{-10}$ deletions per day, per base-pair; $P = 0.0008$) (Fig. 4a, b). We detected no systematic deviation in INDEL mutation rates per chromosome (Supplementary Fig. 6a), and we did not find any enrichment of INDELs at regions associated with common recurrent change in human PSC (for example chromosomes 1q, 12p, 17q, 20q). Due to the overall low number of INDEL mutations, it was not possible to make further meaningful analysis of rates within different regions of the genome.

All growth-condition groups showed some evidence of larger structural rearrangements. Using the BRASS structural rearrangement-calling algorithm (www.github.com/cancerit/BRASS), we detected, in different subclones, three deletions, del (2)(p16.1p16.1), del(10)(q26.13q26.13) and del(12)(q22q22), two tandem duplications, dup(4)(q22.2q22.2), dup(12)(q14.3q14.3) and one translocation,?der(20)t(11;20)(q21;p11.23). In addition, we found one translocation,?der(12)t(8;12)(q21.11;p11.22), in two subclones of one cohort of MShef11 under normal culture conditions, most likely reflecting a mutation early in the culture of the parent clone (Supplementary Table 1). Of these variants, one translocation involved chromosome 12, and another involved chromosome 20, a deletion involved chromosome 10 and a tandem duplication and a deletion involved chromosome 12. These chromosomes are all well known to be associated with common karyotypic changes in human PSC but, with the exception of a breakpoint in a region of chromosome 10p associated with repeated deletions[2], none of the break points involved in the variants observed here were in regions associated with the common karyotypic changes of these chromosomes. However, there were too few such rearrangements to permit a meaningful analysis.

**De novo mutations in genes**. Across all subclones, we detected 5695 de novo SNVs in 4095 genes in the GRCh37 genome

assembly (including those unannotated) (Supplementary Data 1) 4694 mutations mapped to introns and 225 SNVs occured in exons resulting in 90 missense, 7 nonsense and 27 synonymous ammino accid substitutions. A further 985 SNVs mapped to regions within 1 kb up- or downstream of genes, 114 to 5' and 3' untranslated regions (UTRs), 27 to genes encoding non-coding RNAs, and 15 to splice regions or essential splice sites (Supplementary Fig. 6b). We further analysed whether nonsense mutations or mutations in splice sites had any effects on transcript levels (Supplementary Table 2). In only two of these mutations did we find evidence of reduced mRNA levels. MShef11 + Y27632 subclone K2 harbored a nonsense mutation in the gene *DNAJC6*, which exhibited a 43% decrease in expression compared to the mean of other subclones in this cohort (2.7 vs. 4.7) while MShef4 subclone J13 harbored a splice region mutation in the gene *C11orf73*, which exhibited a 32% decrease in expression compared to the mean of its cohort (8.4 vs. 12.34). Eight genes acquired INDELs that produced frame-shift mutations, each occurring in a unique subclone (Supplementary Data 3). Although some of the affected genes are expressed in the undifferentiated cells (Supplementary Data 4), we did not note any that have any obvious special significance for human PSC biology.

We did not find any de novo mutations in genes known to be important in maintaining stem cell pluripotency and homeostasis, nor any mutations in *BCL2L1*, which is frequently amplified in human PSC cultures, or the common 20q11.21 copy-number variation that results in its duplication. A recent study identified a subset of human PSC lines harbouring multiple mutations in the *TP53* gene, which are frequently seen in cancers and which may drive selective advantage in vitro[12]. However, we did not detect any *TP53* point mutations nor INDELs in any of our 80 independent subclones.

We also compared our data to the NCBI public archive of clinically relevant variants (ClinVar) and found a single de novo SNP in one MShef4 subclone, a C > T transition in the penultimate 3' exon of *SNCAIP*—an α-synuclein-interacting protein on chromosome 5q, associated with Parkinson's disease. This substitution has an allelic frequency of 0.00001647, based on 60,706 individuals of diverse genetic backgrounds in the Broad Institute Exome Aggregation consortium. It results in a missense-coding mutation R806C but is not classed as clinically pathogenic. It lies immediately 5' to a second, clinically recognised germline missense-coding SNP also considered to be clinically benign. Overall, the absence of clinically recognised mutations in our data is reassuring, as it suggests that even following prolonged culture the acquisition of known, pathological mutations is extremely low.

**Properties of mutated genes**. We searched for characteristics of genes that had acquired de novo SNVs, in order to understand what might render them more, or less susceptible to mutation (Supplementary Data 4). For this analysis, we defined a gene as the entire annotated region from the GRCh37 genome assembly, comprising introns, exons and UTRs. To measure the relationship between gene size and mutation, we binned genes into 20 kb size bins, and calculated the mean mutational load (SNVs per base-pair of gene) per bin. In a model of stochastic mutation, we would predict an equal mutational load across all genes. However, we observed mutational load decreasing with increasing size, in genes up to 20 kb (Fig. 5a) indicating that the smallest genes were disproportionately mutated. Further, the size distribution of the top 10% most-highly mutated genes was significantly smaller than that of the bottom 10% of their least-mutated counterparts (Fig. 5b). GO analysis using the PANTHER (Protein ANalysis THrough Evolutionary Relationships;

(http://www.pantherdb.org/). Classification System highlighted some association of these genes with various signalling pathways, but as the mutations occurred across independent subclones and growth conditions, we cannot ascribe any particular significance to these enrichments (Supplementary Table 3) (Supplementary Data 5). From these data we conclude that mutation does not occur stochastically over genes, and that other mechanisms must contribute to their mutability.

Active transcription has been identified as a bottleneck for genomic stability, as DNA replication, repair, and transcriptional machinery can physically interfere with one another (reviewed in ref. [27]). Further, local differences in chromatin structure may render those regions of a gene more or less susceptible to mutation. To test the relationship between transcription, gene expression, and mutation in our data we measured the proportion of genes harbouring mutation at varying distance from their transcription start sites (TSS). We found the prevalence of mutation increased with increasing distance from TSS (Fig. 5c), as has been observed previously[24].

To analyse the relationship between gene expression and mutation we used the gene expression estimates from RNA-seq data for MShef4 and MShef11 subclones derived from each growth-condition (Supplementary Data 4). We calculated FPKM expression values of mutated genes in each cell line and growth condition independently, and excluded those genes with log (FPKM) values below 1, considering them to be unexpressed. We then correlated the expression levels of mutated genes with their corresponding mutational load (mutations per bp) and found a trend of positive correlation between gene expression and mutational load (Fig. 5d). To further quantify this effect, we used a Linear Mixed Effect model (LMER) using the formula: *Mutation_Load ~ Gene_Expression + Mutation_Effect + (1 | Gene_Length)*. This accommodates the non-random effect of the gene length as described above and the impact of that mutation type can have on gene expression. The LMER reached convergence and confirmed the positive trend as pictured in the scatter plots; we also found that the LMER applied to mutated genes that were not expressed or with very low expression, described an opposite correlation trend (Supplementary Figure 7).

**Epigenetic changes across the genome**. Epigenetic changes have been detected in human PSC, although evidence that they may provide growth advantages is less clear. Loss of X-inactivation, termed erosion, has been reported a number of times in female human PSC[28,29] and may correlate with the observation that gain of X chromosomes may provide a growth advantage to human PSC as it is one of the non-random genetic changes commonly seen[2,4]. To look for epigenetic changes, we used whole-genome bisulphite sequencing (WGBS) data obtained from the original starting cultures of MShef4 and MShef11 grown in standard conditions, the parental clones derived from those cultures and expanded in different conditions, and the subclones derived from those parental clones (Fig. 1). We measured DNA methylation levels over a range of genomic features annotated in the GRCh38 genome assembly: tiled regions containing 100 CpGs across the entire genome, CpG islands, promoters, and annotated genic regions (Supplementary Data 6–9). We then compared cell lines and growth-conditions at different stages of the experiment.

The original starting populations of MShef4 and MShef11 grown in standard conditions (Fig. 1) had comparable methylation levels to one another across all genomic features (Supplementary Fig. 8). In most cases, the parental clones derived from those starting cultures also had similar methylation patterns to one another. However, Wilcoxon rank-sum testing showed that

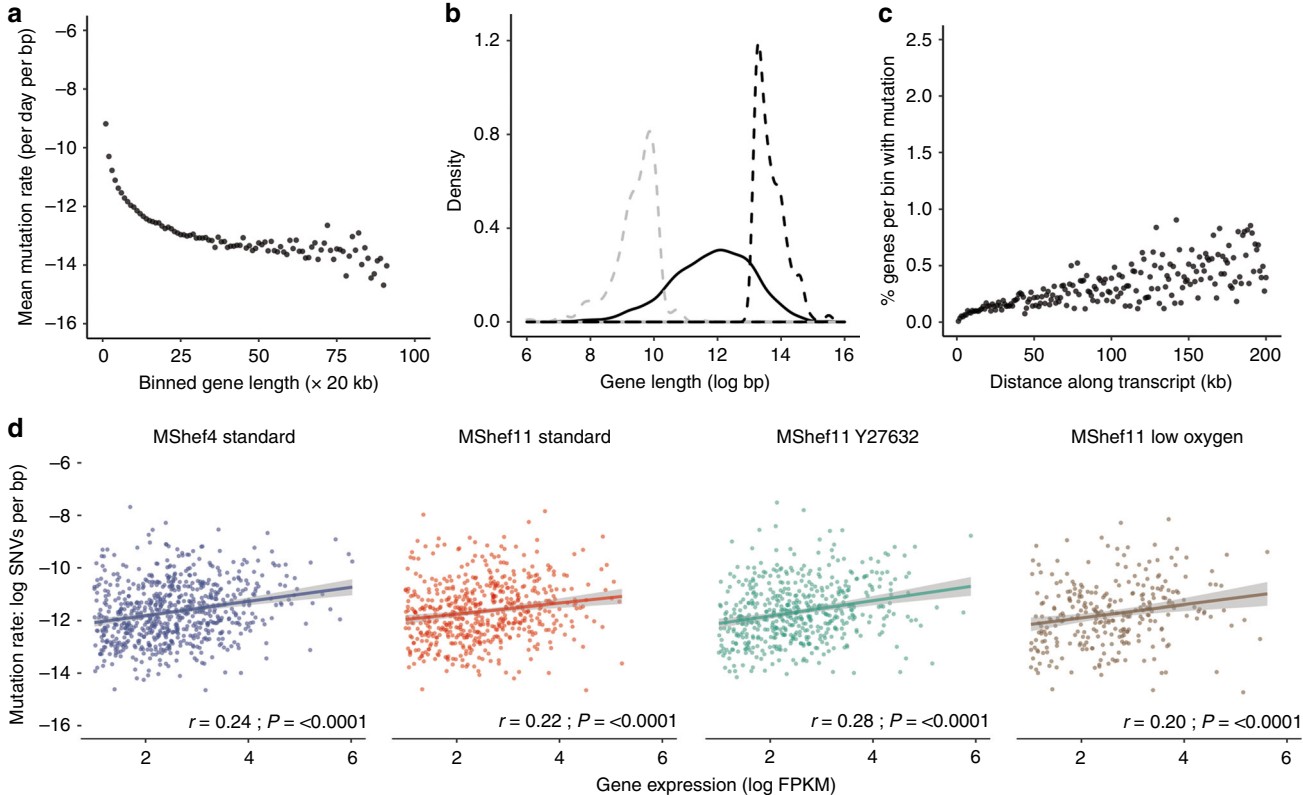

**Fig. 5 Relationship between gene size, expression, and mutation. a** Dot plot showing the mutational load of genes in relation to their size. Mutated genes from all growth-condition groups were binned by size into 20 kb bins. Dots represent individual bins containing genes of increasing size; the *y*-axis indicates the mean number of mutations per base-pair for each bin (natural log scale). Mutational load decreased with increasing gene size. **b** Density plot showing the distribution in gene size for the 10% of genes with highest mutation load (dashed grey line) compared to an equal number of the least-mutated genes (dashed black line) and to all mutated genes (solid black line). Genes with a high mutational burden were small compared to the overall size distribution and were significantly smaller than genes with a low mutational burden. **c** Dot plots showing the percentage of mutated genes per subclone with occurrences of mutation at increasing distances from transcription start sites (TSS). The *x*-axis denotes distance from the TSS (kb); the *y*-axis shows the percentage of genes within each bin that harbour mutation. The data show increasing mutation prevalence with increased distance from TSS. **d** Scatter plots showing of the relationship between mutational load (*y*-axis) and gene expression (*x*-axis) in MShef4 and MShef11 under all conditions (natural log scale). Plots indicate a weak positive correlation between expression and mutation. Solid coloured lines indicate the fitted linear regression models for each group; shaded regions around regression lines indicate 95% confidence intervals for the models. Spearman correlation test statistics and *P*-values are shown for each group (MShef4 standard:.rho = 0.24; MShef11 standard: rho = 0.22; MShef11 + Y27632: rho = 0.28; MShef11 low oxygen: rho = 0.2). Source data are provided in Supplementary Data 4 and 5.

both parental clones grown in low oxygen (G8 and G2) had a significant decrease in methylation compared to those grown in standard conditions (both $P = <0.001$), with a particularly large decrease in G8 and a mean reduction of 3.6% (Fig. 6a). The reduction in methylation seen under low oxygen appeared reversible, as the subclones derived from G2 and G8 in low oxygen but expanded under standard conditions prior to WGBS showed reversion back to 91% methylation, equivalent to their counterparts from standard conditions. (Fig. 6a). Overall we found a weak relationship between promoter methylation and gene expression.

MShef4 subclones (J1-20) as a group showed striking hypermethylation of CpG island-containing (CGI) promoters compared to MShef11 subclones (Supplementary Fig. 8). To test if this effect was due to a single aberrant subclone skewing the grouped data, or if the hypermethylation had occurred in many independent subclones, we used higher-depth WGBS data from a subset of seven independent MShef4 subclones (Supplementary Data 10, 11) and compared the methylation levels of each to their parental clone, B8. Each subclone showed substantial hypermethylation of a subset of CpG island-containing promoters (Fig. 6b, top panels; Supplementary Fig. 9, left panels). This

showed that the effect was not restricted to a single aberrant subclone and had likely occurred in all subclones in the cohort following single-cell deposition and expansion. Of the 11,677 CGI promoters measured, 1905 were hypermethylated by at least 20% across all seven subclones (Supplementary Data 12) although the levels were highly variable. However, only two CGI-containing promoters were hypermethylated by an equivalent level (±5%) across all subclones (these were directly upstream of the long non-coding RNA *FP236383.12* and the ribosomal gene *RNA5-8S5*, located on chromosomes 21 and 22, respectively) indicating no consistency in the patterns of hypermethylation and no common predecessor with the epigenetic alterations. In contrast, methylation of non-CGI promoters was similar between each subclone and its parental clone, B8 (Fig. 6b, bottom panels; Supplementary Fig. 9, right panels). We measured the expression of de novo methyltransferase genes (DNMTs) in MShef4 subclones (Supplementary Data 13) and found significantly elevated expression of *DNMT3B* and *DNMT3L* ($P = 7.2^{-8}$; $P = 2^{-8}$) compared to MShef11 subclones (Fig. 6c). Both genes are involved in de novo methylation, and their relative overexpression in MShef4 subclones may account for the observed hypermethylation of CpG island-containing promoters.

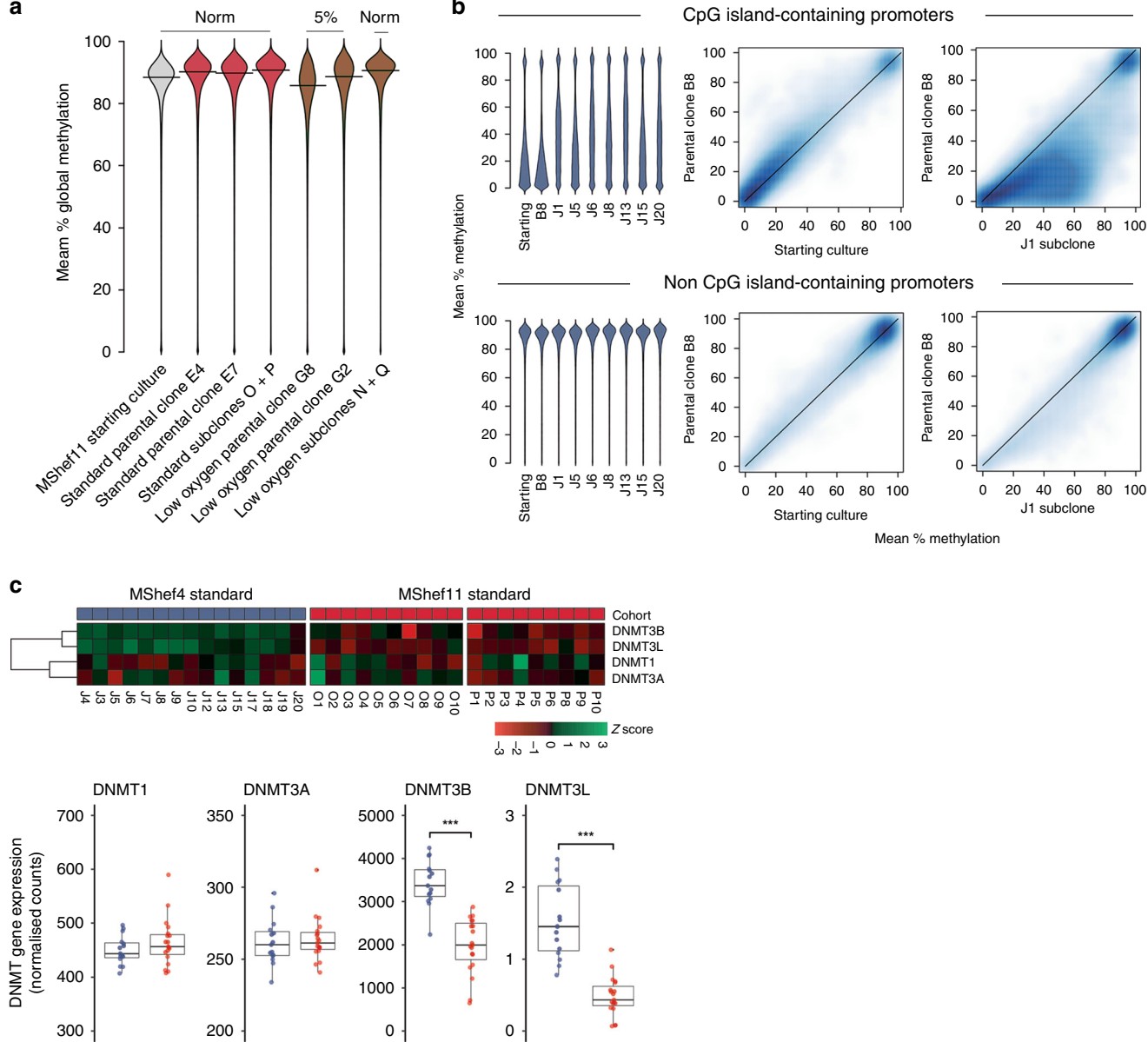

**Fig. 6 Epigenetic changes. a** Beanplots showing global DNA methylation levels of MShef11 starting cultures, parental clones grown in standard or low oxygen conditions, and the subclones derived from those conditions. Methylation was assessed over tiled probes across the genome, each containing 100 CpGs. Both parental clones grown in low oxygen show a small decrease in methylation, particularly clone G8. This decrease in methylation was reversible, as subclones derived from low oxygen revert to normal methylation levels following expansion in standard conditions. Source data are provided in Supplementary Data 6. **b** DNA methylation of promoters in MShef4 subclones (N = 20). Promoter regions were defined as regions surrounding transcription start sites, with 1500 kb upstream and 500 kb downstream context. Beanplots (left panels) show DNA methylation levels of promoters with and without CpG islands, in a randomly selected subset of MShef4 J subclones. For both classes of promoter, the starting culture and the parental clone, B8, derived from that starting culture had comparable methylation levels, whereas each J subclone showed hypermethylation of a subset of CpG island-containing promoters compared to its parental clone, B8. Density scatter plots (right panels) show the distribution of promoter methylation levels between the MShef4 starting culture and its derived parental clone, B8, and between the parental clone B8 and an example subclone, J1. The density scatter plots show comparable levels of non-CpG island-promoter methylation between starting culture material and parental clone B8, but hypermethylation of CpG island-containing promoters in subclone J1 compared to parental clone B8. Source data are provided in Supplementary Data 9–12. **c** Top panel; heatmap showing the relative expression of de novo *methyltransferase* (*DNMT*) genes in MShef4 (N = 20) and MShef11 (N = 19) subclones from standard conditions. Lower panel; box and whisker plots showing the normalised expression counts (fragments per million reads) for *DNMT* genes in MShef4 and MShef11 standard subclones. Both *DNMT3B* and *DNMT3L* are more highly expressed in MShef4 subclones ($P = 7.2^{-8}$ and $P = 2^{-8}$). Boxes represent the 25th–75th percentiles of the data; whiskers show the min and max range of the data; horizontal lines indicate the median value. ns: $P > 0.05$; *$P \leq 0.05$; **$P \leq 0.01$, ***$P \leq 0.001$; ****$P \leq 0.0001$. Source data are provided in Supplementary Data 13.

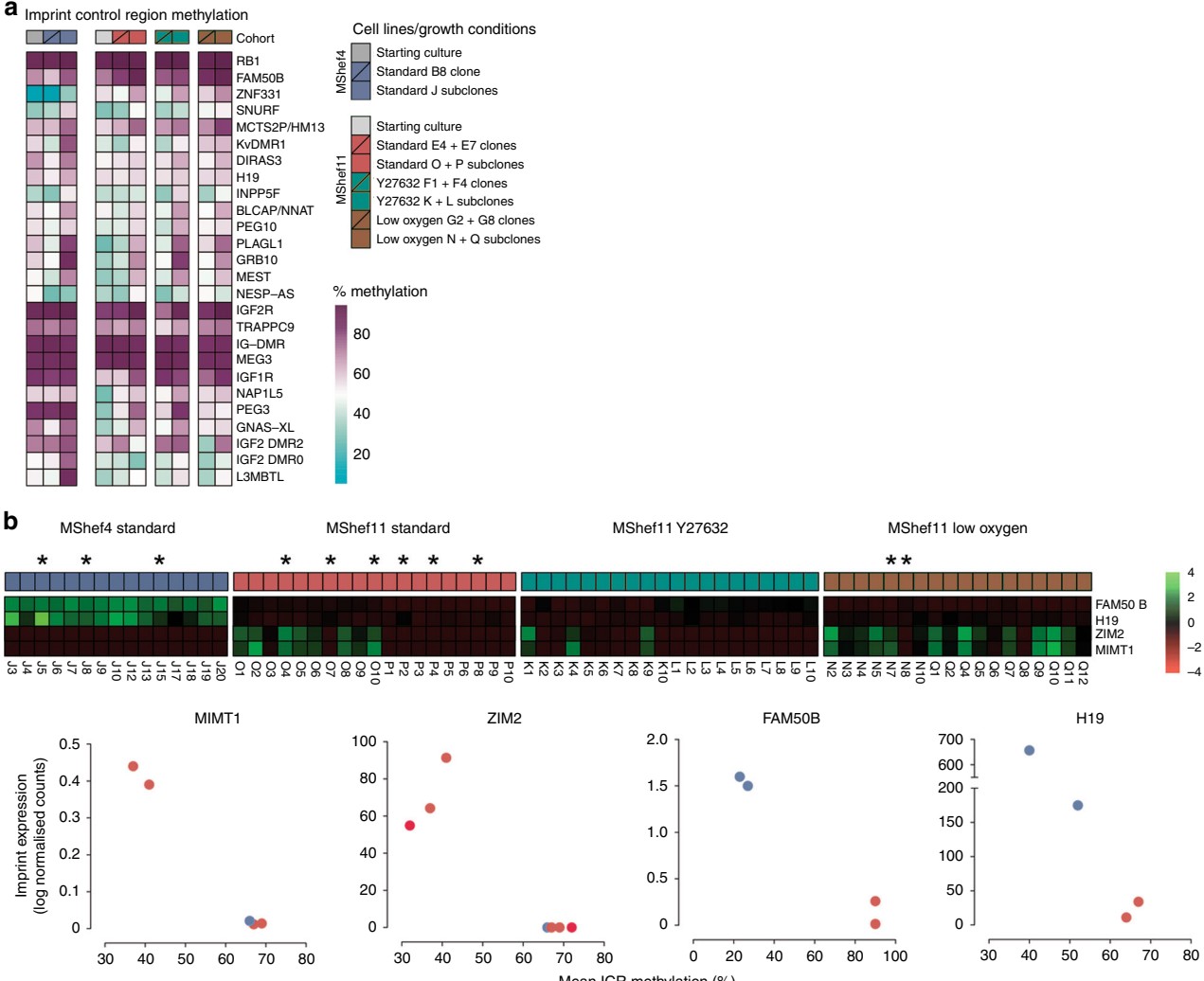

**Fig. 7 Imprint methylation and expression. a** Heatmap showing the methylation levels of 26 imprint control regions (ICRs) in starting cultures, parental clones, and subclones, grouped by cell line and growth-condition. Individual ICRs show a wide range of methylation levels with some notable inter- and intra-group variation. Source data are provided in Supplementary Data 14. **b** Top panel; heatmap showing the relative expression of four selected imprinted genes: *FAM50B, H19, ZIM2,* and *MIMT1* in MShef4 and MShef11 subclones. Both *FAM50B* and *H19* are more highly expressed in MShef4 subclones; *ZIM2* and *MIMT1* are more variable in their expression. Lower panel; dot plots showing the correlation between ICR methylation and the expression of four imprinted genes, in a subset of MShef4 and MShef11 subclones with WGBS data of sufficient depth to permit such analysis (selected subclones indicated by asterisks in top panel). In all cases differences in gene expression correlate with a change in ICR methylation. Source data are provided in Supplementary Data 15.

**Differences in imprint methylation and expression**. Given the epigenetic differences observed in our data, we examined imprinting in the subclones, which is regulated by the epigenetic status of imprint control regions (ICRs). Although imprinting in human PSC seems to be relatively stable, loss of imprints are sometimes seen, though with unknown significance[30]. In contrast, naive human PSC[31,32] have less stable imprinting. For this analysis, we integrated WGBS and RNA-seq data to look for differences in ICR methylation and related imprinted gene expression (Supplementary Data 14). Comparing the methylation status of 26 known ICRs in the starting cultures, in the parental clones grown in different growth-conditions, and in their respective subclones, the mean ICR methylation was 64% across all groups, and close to the expected 50% CpG methylation resulting from one fully methylated and one unmethylated parental allele. Individual ICRs ranged between 7.6–96.9%, and methylation of some ICRs fluctuated between groups (Fig. 7a). Closer examination showed differences in the overall methylation

of ICRs, between starting cultures, the parental clones and the subclones. The MShef4 starting culture had higher average ICR methylation than its MShef11 counterpart; MShef11 clones grown in low oxygen had elevated ICR methylation compared to their parental culture, and all subclone groups acquired higher ICR methylation compared to their clones-of-origin (Supplementary Fig. 10).

To examine the effects of differences in ICR methylation, we measured the expression of 75 imprinted genes in our subclones and compared cell lines and growth-condition groups (Supplementary Data 14). All MShef4 subclones showed differential expression of several clusters of imprints, relative to MShef11 (Supplementary Fig. 11) with one cluster having comparatively increased expression, and at least two clusters showing reduced expression. In particular, expression of *H19, IGF2,* and *NLRP2* was systematically different between MShef4 and MShef11 subclones. Furthermore, imprint expression between MShef4 subclones was more homogenous than

between MShef11 subclones, which showed greater variation within, and between growth-condition groups (Supplementary Fig. 12). In a set of subclones that showed significant variation in imprint expression, and for which sufficient WGBS depth was available, four imprinted genes, *MIMT1*, *ZIM2*, *FAM50B* and *H19*, all showed an inverse correlation between ICR methylation and gene expression as anticipated (Fig. 7b). Overall, the methylation patterns were subject to environmental growth conditions, in a way that was cell-line dependent.

## Discussion

Understanding genetic and epigenetic change in human PSC during in vitro culture is important in the context of regenerative medicine, as these cells represent a starting point for stem cell-based therapeutics. Thus, developing culture conditions that minimise change is important for improving safety in clinical applications of human PSC-derived therapeutics. It is encouraging that following culture over an extended period of time and multiple passages, both the human PSC in this study acquired a relatively low burden of single-nucleotide base substitutions, and none of the common genetic variation seen in human PSC. This is consistent with the view that the common non-random genetic changes observed in human PSC are driven by the selective growth advantage of rare but random mutations. However, estimating the underlying rate of mutation in PSC is difficult because by the time mutants become detectable their frequency may be grossly distorted by the effects of selection of those that offer a growth advantage. The clongenic strategy that we adopted to obviate this difficulty is, however, complex and expensive, so that it was only feasible to analyse two human ES cell lines, and the necessary production and selection of clones with which to carry out the analysis might itself have introduced distortions. Nevertheless, the rates we observed under normal growth conditions ($0.37 \times 10^{-9}$ and $0.28 \times 10^{-9}$ SNV per day, for genetically independent human ES cell lines (MShef4 and MShef11, respectively) are not significantly different and are comparable with two other recent studies of human PSC ($0.18 \times 10^{-9}$ and $1 \times 10^{-9}$ SNV per base-pair, per cell division, respectively)[3,18]. They are substantially lower than the mutation rates estimated in somatic cells. The two ES cell lines used in our present study are mostly likely genetically unrelated, and one was derived from a frozen embryo and one from a fresh embryo. By contrast the lines studied by Rouhani were iPS cell lines, derived by reprogramming. That the mutation rate in all of these lines was comparably low, suggests that this may be a feature of PSC in general, irrespective of their means of derivation. It is also notable that in a study of a single locus, *Aprt*, in mouse ES cells,[33] concluded that the mutation rate in mouse ES cells is substantially lower than in corresponding somatic cells.

An increasing number of reports link culture conditions and reagents to the occurrence of genetic variants[7,34–36]. Clearly, existing in vitro culture conditions are sub-optimal for the maintenance of genetically normal stem cells and, while the mechanisms that underlie genetic variation are unclear, it is possible that exposure to factors such as mutagenic chemical components in growth media, and the accumulation of reactive oxygen species could drive mutation. Importantly, though, the Rho-Kinase inhibitor, Y27632, which is now widely used in PSC culture did not affect the mutation rate despite concerns that its anti-apoptotic effects might enhance the appearance of mutations. However, our data did show about a 50% reduction in mutation rate in cells cultured under low (5%) oxygen compared to standard conditions, which may be a consequence of reduced levels of reactive oxygen species generated through oxidative stress. This oxygen level, which has also been reported to affect

the behaviour and differentiation capacity of PSC (e.g.[37,38]), more closely mimics the oxygen tension experienced in the early embryo[39]. Our results do suggest a benefit for maintaining human PSC cultures in a low oxygen environment, though to avoid exposure to fluctuating oxygen levels, for example to atmospheric oxygen during passaging and other manipulations, this requires the use of a dedicated low oxygen workstation, as we have done in this study.

The mechanisms underlying mutation have been investigated most thoroughly in the context of cancer, where the patterns of base substitution observed in specific cancer types has been deconvoluted and linked to their causative environmental agents or to aberrant biological mechanisms[24,25]. In this study, cultures from all growth-conditions showed a strong correlation with a mutation signature frequently detected in cultured cells and linked to oxidative stress (COSMIC signature 18). Similar observations were also made in other studies of the mutational burden in human iPS cells[40–42]. Consistent with this, low oxygen culture induced a change in mutational signature, resulting in a reduced C > A component in signature 18.

A stochastic model of mutation would predict a roughly equal rate of change across all regions of the genome. Indeed, we found that mutation was evenly distributed across the chromosomes, with the exception of the X chromosome, which exhibited a slightly elevated mutation rate compared to the genome-wide average. Because MShef4 and MShef11 are both male cell lines it is possible that this is a male specific phenomenon. For example, the absence of a second copy of the X chromosome for repair by homologous recombination, or increased transcription rates from X chromosomes in male PSC to compensate for the presence of two active X chromosomes in female PSC, may partly account for the elevated mutation rate. Nevertheless, we did find differences between genomic regions, with an elevated mutation rate in intergenic DNA compared to coding-region DNA, though in the genic regions we found no difference in mutation rate between exons and introns. This suggest that differences in chromatin structure or functional activity render those regions more or less mutagenic and that an antagonistic relationship between transcription and DNA repair[27] may be compensated by preferential DNA repair in genic regions, as has been reported by others[22,23]. On the other hand, high transcriptional activity may hinder DNA repair processes as we did find a weak correlation between mutational load and gene activity, a relationship that has previously been observed in yeast[43], though in contrast to observations in tumor cells[24]. That discrepancy might reflect differences in cell cycle control and response to DNA damage in embryonic cells compared to somatic cells[20].

Further to the rates of change caused by mutation, we also observed significant differences in the epigenetic status of the two human ES cell lines we studied. Previous studies of PSC have detected cell-line-specific and culture-induced epigenetic changes in general DNA methylation[34,44] and at specific loci such as imprint control regions[2,30,45]. We detected a small reduction in global DNA methylation levels in MShef11 clones cultured in low oxygen, which was reversible following subcloning and expansion in standard oxygen conditions. In mouse ES cells, resetting to a naive state is accompanied by hypomethylation of the genome to a level comparable with the inner cell mass (ICM)[31,46,47]. Thus, if low oxygen culture mimics conditions in the early embryo, we might expect to see some level of hypomethylation. However, the small decrease in methylation detected in this study (mean reduction of 3.6% in CpG methylation in low oxygen cultures) is much less than the decrease seen in cells extracted from the ICM or following naive reprogramming (reduction of ~40% in CpG methylation in ICM or naïve human ES cells[48]). Furthermore, one clone appeared more responsive to low oxygen than the

other, making it difficult to accurately assess the effect of this condition on DNA methylation. We also saw significant hyper-methylation of CpG island-containing promoters following subcloning of MShef4. This hypermethylation had occurred in multiple independent subclones, suggesting a global effect across the entire cohort. This event might be a stress-induced response to single-cell cloning and is worth considering in the context of PSC maintenance, given the dynamic nature of epigenetic change in PSCs. Finally, we saw a significant variation in the expression of imprinted genes and the methylation status of imprint control regions (ICRs). Overall, we detected variability in ICR methylation in subclones, compared to their parental clones or the starting cultures from which they were derived. MShef4 subclones exhibited a different pattern of imprint expression to MShef11 subclones, with little variation between subclones. For some imprints, MShef11 subclones showed significant subclonal variation, and we were able to detect specific instances in which changes in ICR methylation correlated with changes in accompanying imprint expression. These findings suggest that DNA methylation is highly dynamic, responsive to growth-conditions and culture practices, and can vary in a cell-line-specific manner. However, as we did not find a correlation between epigenetic change and mutation, the two may be unrelated forms of variation.

Overall, the striking conclusion from this study is the low mutation rate in human PSC, whether affecting SNV or INDELS, despite the frequent reports of common genetic variants in the literature. Most likely, the latter reflects an ascertainment bias. In the ISCI study of 120 pairs of human PSC in early and late passage, 79 lines remained karyotypically normal[2] while in a sequencing study of 140 human ES cell lines[12], only six acquired mutations in TP53, all results consistent with an underlying low mutation rate in human PSC. Of course, one unknown is whether PSC lines that have acquired growth advantages through long periods in culture may have an altered mutation rate, perhaps a mutator phenotype. Evidently, the mutation burden in human PSC can be reduced by culture conditions, such in a low oxygen environment, but it seems that the appearance of common variants is largely a consequence of selection rather than underlying mutation. Minimising the appearance of such variants will, then, depend primarily upon identification and moderating of the mechanisms by which they exhibit a growth advantage.

## Methods

**Cell lines and culture methods.** Derivation and maintenance of the MShef4 and MShef11 cell lines was performed in the Sheffield Centre for Stem Cell Biology under HFEA licence R0115-8-A (Centre 0191) and HTA licence 22510, in a clean room setting, following strict standard operating procedures. The embryos used to derive MShef4 (frozen embryo) and MShef11 (fresh embryo) were donated from different Assisted Conception Units, and so likely from different donors, following fully informed consent, with no financial benefit to the donors, and were surplus or unsuitable for their IVF treatment. Briefly, the embryos, were cultured using standard IVF culture media (Medicult), to the blastocyst stage. Following removal of the trophectoderm using a dissection laser the embryos were explanted whole onto either mitotically inactivated human neonatal fibroblasts (human feeders) in standard KOSR/KODMEM (Life Technologies) medium in the case of MShef4 or onto Laminin-511 (Biolamina) and Nutristem medium (Biological Industries) in the case of MShef11. Both cell lines were initially maintained at 37ºC under 5% $O_2$/5% $CO_2$, until the lines were established, after which maintenance switched 5% $CO_2$ in air at 37 °C. Cultures were passaged using a manual technique, cutting selected colonies under a dissection microscope at an average split ratio of 1:2 every 7 days. Both lines have been deposited at the UK Stem Cell bank.

Parental material for these experiments was taken from fully characterised research bank frozen stocks known to contain high quality undifferentiated human ES cells, with a normal 46, XY karyotype frozen at passage P36 (MShef4) and P15 (MShef11). Cells were thawed onto mitomycin C-inactivated human fibroblast feeders, maintained in Nutristem and passaged using the manual cutting technique. After expansion for seven (MShef4) and five (MShef11) passages, the lines were cloned by single-cell deposition using a FACSJazz flow cytometer (Becton Dickinson) into 96-well plates with human feeders, Nutristem medium and

Rho-Kinase (ROCK) inhibitor, Y27632[49]. After 24 h the medium was changed to remove ROCK inhibitor and the developing clones observed over 2 weeks before being manually picked and expanded in Nutristem on human feeders at 37 °C under an atmosphere of 5% $CO_2$ in air, without the use of Y27632. From each line we selected a clone that we then recloned by single-cell deposition as above to yield parent clones at passage levels P53 (MShef4) and P35 (MShef11) (Fig. 1) with which to initiate the mutation analysis.

In the case of MShef4, we initially isolated ten parent clones by single-cell deposition. At the earliest possible time during their expansion we carried out an initial cytogenetic and SNP array screen to exclude any clones that may have acquired gross karyotypic changes (Supplementary Table 4). A single parent clone (designated B8) was selected for mutation analysis. This was grown under standard growth-conditions (Nutristem medium and mitomycin C treated human fibroblast feeders without the use of Y27632) for a total of 109 days (19 passages) following single-cell deposition, after which a cohort of 20 subclones (Cohort J) was isolated and expanded through a further five passages to provide sufficient DNA and RNA for sequencing.

In the case of MShef11, a number of parent clones were similarly isolated but then grown under one of three conditions: Ten clones were cultured under the same standard conditions as the MShef4 clones, 10 clones were cultured under similar conditions but with the ROCK inhibitor, Y27632, added during each passage and 10 clones, to be maintained in low oxygen, were immediately transferred after single-cell deposition to a Ruskinn Sci-tive Workstation (Ruskinn Technology, Bridgend, Wales), equipped with an integrated microscope and incubator so that the cells could be passaged continually under an atmosphere of 5% $O_2$/5% $CO_2$ in nitrogen without exposure to atmospheric oxygen. These clones were expanded and screened as early as possible by cytogenetics and SNP array to exclude gross variants, and genetically normal clones were carried forward for further analysis (Supplementary Table 4). Two such parent clones (E4 and E7) were grown under standard conditions, and two (F1 and F4) were grown in the presence of the Rho-Kinase inhibitor at each passage, for 111 days (25 passages) and 115 days (25 passages), respectively. We then produced two cohorts of subclones from each parent clone for sequence analysis: Standard condition, Cohorts O and P (10 and 9 subclones, respectively); Rho-Kinase inhibitor condition, Cohorts K and L (10 subclones each). In the case of low oxygen, two parent clones (G8 and G2) were grown for 111 days (25 passages) but for technical reasons we were only able to produce nine viable subclones (Cohort N) from one of these (clone G8). Therefore, a further cohort of 12 subclones (Cohort Q) was derived from a continuation culture of the first low oxygen clone, G8, but only after it had been removed from low oxygen at 111 days and maintained under standard conditions for a further 28 days (six passages) (Fig. 1).

**DNA sequencing.** Genomic DNA was isolated (DNeasy kit, Qiagen) from each parental cell line, all clones and subclones and used for whole-genome sequencing[3] using the Illumina HiSeqX10 platform (Supplementary Data 15). For whole-genome bisulphite (WGBS), libraries[50] were prepared by sonicating 500 ng genomic DNA using a Covaris Sonicator into 300-400 bp long fragments, followed by end-repair, A-tailing and methylated adapter (Illumina) ligation using NEB-Next reagents. Subsequently, libraries were bisulphite-treated using EZ DNA Methylation-Direct Kit (Zymo), followed by library amplification with indexed primers using KAPA HiFi Uracil+ HotStart DNA Polymerase (Roche). All amplified libraries were purified using AMPure XP beads (Agencourt) and assessed for quality and quantity using High-Sensitivity DNA chips on the Agilent Bioanalyzer. High-throughput sequencing of all libraries was carried out with a 125 bp paired-end protocol on a HiSeq 2000 instrument (Illumina). Raw sequence reads from WGBS libraries were trimmed to remove poor quality reads and adapter contamination, using Trim Galore (v0.4.4). The remaining sequences were mapped using Bismark (v0.18.0)[51] with default parameters to the human reference genome GRCh38 in paired-end mode. Reads were deduplicated and CpG methylation calls were extracted from the deduplicated mapping output using the Bismark methylation extractor (v0.18.0) in paired-end mode. Clones grown in different conditions were sequenced to 15–23X genomic coverage. Subclones were initially multiplexed by group, resulting in 0.8X average coverage per subclone, and 16X average coverage per group. Selected subclones were individually re-sequenced, resulting in 10X average coverage per subclone (Supplementary Data 16).

**RNA sequencing.** Total RNA was isolated using an RNeasy kit (Qiagen). Poly-A-enriched libraries were generated at the Wellcome Trust Sanger Institute. RNA-seq libraries were sequenced on Illumina HiSeq2500 by 75-bp paired-end sequencing. Raw sequence reads were trimmed to remove poor quality reads and adaptor contamination using Trim Galore and mapped to the human GRCh37 hs37d5 reference assembly accompanied by the Ensembl 75 human transcriptome annotations with HiSAT2 (https://www.nature.com/articles/nmeth.3317). On average, 24X exonic coverage was obtained per subclone group (Supplementary Data 16).

**Data processing and quality control.** For WGS data, 150 bp paired-end trimmed sequencing reads were aligned to the GRCh37 reference genome using BWA. Marking of duplicates and their removal were performed using biobambam (www.github.com/gt1/biobambam2). Mutations were then called using the CaVEMan

(Cancer Variants Through Expectation Maximization) computational framework[25]. All somatic mutations that passed the standard threshold established by Alexandrov et al.[25] were filtered for Median Alignment Score (ASMD) of reads greater than 130, and no clipped bases were considered. This identified SNVs present in each set of subclones from both lines and all culture conditions, by subtracting the genetic variation in the relevant parental line for each subclone. Annotated SNPs from the general population and matching with dbSNP entries, which were present in the parental cultures and clones were also removed. From all 80 subclones from both cell lines and all culture conditions 19,579 de novo SNVs (with respect to genomic position and base substitution) passed the initial quality score criteria described above (Supplementary Figures 1-4; Supplementary Data 1). Of these, we excluded 4069, which fell below an allelic frequency of 0.2. Additionally, we omitted from subsequent analyses one MShef11 subclone from the +Y27632 culture condition, and one MShef11 subclone from the low oxygen culture condition since both exhibited very high mutation rates with unusual mixed allelic frequencies, most likely because they were not derived from single cells. We also excluded from the analysis a set of 2350 SNVs that recurred in two or more subclones since they most likely represented mis-calls. After these eliminations, a set of 12,555 SNVs remained and was used to calculate the mutation rates for each cell line under each growth condition. In the case of INDELs we used the *PINDEL* INDEL-calling algorithm (www.github.com/genome/pindel) to detect mutation; we only considered those that passed a standard filtering score >250. From all 80 subclones from both cell lines and all culture conditions 1884 INDELs passed the initial quality score criteria described above (Supplementary Figs. 1-4; Supplementary Data 3). Of these, we excluded 713 that fell below an allelic frequency of 0.2 or the base length was greater than 100. We also excluded the two outlier subclones as previously. After these eliminations, a set of 1171 INDELs remained and was used to calculate the mutation rates for each cell line under each growth condition.

RNA-seq data were analysed using the freely available sequence data analysis software SeqMonk v1.38.2 (Andrews S; Babraham Bioinformatics, https://www.bioinformatics.babraham.ac.uk/projects/seqmonk/). QC was performed using the built-in SeqMonk RNA QC pipeline to assess data content and mapping efficiency. Genome features were derived from the GRCh37 genome assembly in SeqMonk. For inter-sample gene expression comparisons, data were quantified as non-transcript-length-normalised counts (fragments per million reads) of merged transcript isoforms, using the SeqMonk RNA-seq quantification pipeline. For analysis of gene expression within a single sample, data were quantified as transcript length-normalised counts (fragments per kilobase per million reads) of merged transcript isoforms, using the SeqMonk RNA-seq quantification pipeline.

Bisulphite data was also analysed using SeqMonk v1.38.2 (Andrews S; Babraham Bioinformatics, https://www.bioinformatics.babraham.ac.uk/projects/seqmonk/). Quality control was performed within SeqMonk to assess average read coverage and depth over features of interest. Duplicate parental clones cultured in different growth-conditions were analysed as replicate sets; multiple subclones derived from each condition were analysed as grouped data. Analysis was performed using built-in bisulphite quantification pipelines, with the stringency of analysis matched to the level and depth of coverage of samples.

**Mutation signatures**. All selected mutations were then used to extract mutational signatures. For this purpose, the matrix decomposition algorithm was used as implemented in the R package *SomaticSignatures*[52]. *SomaticSignatures* identifies mutational signatures using the methodology described by Nik-Zainal[24]. Signatures identified were then compared with the set of somatic mutations identified by Alexandrov et al.[25] using *deconstructSig* and *mutationalPatterns* packages in R. Cosine similarity was used to rank the signature against the catalogue.

**Rate calculations and statistical analysis**. Genome mutation rates were calculated by dividing the number of mutations per replicate subclone by the number of days maintained in culture, then by the total number of sequenced base-pairs in the GRCh37 genome assembly[21]. Mutation rates of individual chromosomes or selected genomic features were calculated by dividing the number of mutations per subclone by the number of days in culture, then by the total base-pair content of each chromosome or genomic feature. In all calculations, rates were normalised to account for two copies of each autosome.

To test for differences between the mutation rates of growth-condition groups, we first used D'Agostino & Pearson and Shapiro-Wilk normality tests to measure the distribution of the data. Data that passed both tests at $P = < 0.05$ were considered normally distributed. When data were normally distributed, we used Student's t-test with Welch's correction to correct for unequal standard deviations. When one or both sets of data were not normally distributed, we used non-parametric Mann-Whitney rank testing.

To identify chromosomes with mutation rates that varied significantly from the genome-wide rate, we first used D'Agostino & Pearson and Shapiro-Wilk normality tests to measure the distribution of the data for each chromosome and for the genome-wide rate. Data that passed both tests at $P = < 0.05$ were considered normally distributed. In each growth-condition individual chromosomes were compared with the respective genome-wide rate on a pairwise basis. When both the chromosomal and genome-wide data were normally distributed, we compared them using Student's t-test with Welch's correction to correct for unequal standard deviations. When one or both of the chromosomal or genome-wide data were not normally distributed we used non-parametric Mann-Whitney rank testing. Data were corrected for multiple tests (22 tests x 4 growth conditions = 88 tests) using the Bonferroni method to produce adjusted P-values.

Intronic and exonic regions were classified based on their annotated positions in the GRCh37 genome assembly. Intergenic DNA was classified as regions between annotated genic regions and was mutually exclusive to the exonic and intronic regions. Unsequenced regions of intergenic DNA (including the ends of some chromosomes, and at centromeric regions) were excluded from the analysis. To calculate mutation rates the number of SNVs falling in each region was divided by the total genomic DNA content of those regions, accounting for four copies of each autosome. Other genomic regions were identified according to annotation tracks from the Ensembl and ENCODE (https://www.encodeproject.org/reference-epigenomes/ENCSR323FKB/) websites.

For mutations falling within annotated genes, the distance of each SNVs relative to the TSS was calculated. TSS were identified based on annotations from the UCSC table browser. Mutations in genes were then binned into 1000 bp bins relative to their respective TSS, with increasing distance from the TSS. For each bin, the proportion of genes with a mutation at that distance from the TSS was calculated with relative to the total number of genes per bin, based on all those in the GRCh37 genome assembly.

FPKM values for mutated genes were calculated using SeqMonk software and combined with mutation rate data derived from CaVEMan. Log transformed values were checked for normality of distribution in Prism 6, then correlated by Spearman Product Moment testing using the base R function *cor.test* and the in-built Prism 6 correlation test. The LMER was implemented using the function *lmer* () in R within the package *lme4*.

**Reporting summary**. Further information on research design is available in the Nature Research Reporting Summary linked to this article.

## Data availability

The authors declare that all data supporting the findings of this study are available within the article and its supplementary information files or from the corresponding author upon reasonable request. All raw data have been deposited in the European Genome-phenome Archive under accession codes: EGAS00001001561 (whole Genome sequencing), EGAS00001001625 (whole Genome Bisulphite Sequencing) and EGAS00001001655 (mRNA sequencing). All processed source data underlying all Figures and Supplementary Figures and Tables are available in the Supplementary Data files as indicated in the relevant Figure Legends.

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

## Acknowledgements

This work was funded by the UK Regenerative Medicine Platform under a grant (MR/L012537/1) from the MRC, and also supported by the Wellcome Trust (WT206194)

## Author contributions

The scientific conception, management and co-ordination of the project was provided by P.W.A. and K.Y.; The derivation and banking of the MShef4 and MShef11 human ES cell lines was carried out by Z.H., R.W. and A.W. under the direction of H.D.M. Culture and cloning of the cells for mutation analysis and DNA, RNA and library preparation was carried out by S.G., Z.H., O.T., R.W. and A.W.; Analysis of whole-genome and bisuphite DNA sequence data and RNA sequence data were carried out by J.A., S.A., F.K., F.v.M., M.M., S.N.-Z., W.R. and O.T.; The manuscript was drafted by P.W.A., I.B., P.J.G., Z.H., F.v.M., M.M., S.N.-Z., W.R., O.T. and K.Y.

## Competing interests

W.R. is a consultant and shareholder of Cambridge Epigenetix but all other authors declare no competing interests.
