## [Peer Review File · Nature Communications]

Reviewers' Comments:

Reviewer #1:

Remarks to the Author:

Thompson and colleagues have studied the genetic and epigenetic changes occurring during in vitro culture of two clinical grade human embryonic stem cell (hESC) lines, and provided an important data to assess the safety of clinical application of hESC. The authors have generated a large number of datasets, and performed a thorough investigation to identify different types of mutations in hESC such as point mutations, insertions, deletions, and chromosomal translocations. The authors have discovered that the mutation rate was low in hESC during cell culture and that the mutation rate was reduced under low oxygen conditions. Epigenetic analysis has shown that clones grown in low oxygen conditions had elevated methylation compared to their parental controls in one cell line, although the functional consequence of this elevation is not shown. While this study might have limited scientific advancements, the authors have reported important findings to facilitate the translation of hESC to the clinic. However, the generality of the conclusions drawn from this study using only two hESC lines is uncertain. Listed below are some points of concern.

1. The authors have used only two hESC lines to draw their conclusions. The authors can mention in the text how commonly these cell lines are used. Importantly, how general is the conclusion drawn from this study (also considering the last paragraph of the discussion section on page 16, in which 41 lines out of 120 PSCs showed abnormal karyotype in the ISCI study)?

2. The authors found that the mutation rate was higher in intergenic regions when compared to genic regions (Fig. 2c). The authors can perform deeper analysis in the context of epigenetic status, to examine how many mutations overlapped open chromatin and heterochromatin regions in hESC. The authors can overlay different histone marks and transcription factor binding to identify mutations present in active and repressed regions, similar to the analysis performed in Yoshihara et al. 2017 Cell Reports.

3. The authors reported a total of 225 SNVs in exons, including 7 nonsense mutations, and 85 (this number is not shown in the text) non-synonymous mutations. The authors have provided a complete list of mutations with their locations and other relevant details in the Supplementary Data 1. It would be nice if the authors can show the number of missense, nonsense mutations etc. in the form of a bar or pie chart. The authors can also provide more data about the mutated genes. What kind of genes are affected? This would help the readers better understand the data. Similarly, a figure can also be shown for number of mutations seen in 5' UTRs, coding exons, splice sites etc.

4. In Fig. 5b, it would be interesting to see which genes have a high mutational load with short gene lengths and low mutational load with long gene lengths. GO analysis might be also be helpful in this context. Also, the legend states "solid grey line" but I think the authors mean "solid black line".

5. The authors have generated many datasets using whole-genome sequencing (WGS), whole-genome bisulphite sequencing (WGBS) and RNA-seq. The authors have provided read statistics for WGBS and RNA-seq data, they can also provide the same for WGS data. Although the authors have integrated WGBS and RNA-seq data for imprint control regions analysis, this kind of integrative analysis can be expanded. It would be nice if the authors can integrate WGS and RNA-seq data to study the functional effect of mutations on the transcriptome. For example, what effect do nonsense and splice site mutations have on the transcripts? Similarly, the authors can integrate the WGBS and RNA-seq data to understand the effect of differential methylation of CpG promoters (shown in Fig. 6b) on gene expression.

6. The authors state that there was a positive correlation between gene expression and mutational

load (Fig. 5d). What were the correlation coefficients and p values? The authors used log (FPKM) which is just natural log, it may be better to show gene expression values on a log₂ scale.

7. The authors have mapped their sequencing data using two human genome assemblies (GRCh37 and 38). For consistency, it would be best to use one genome assembly.

8. Figs. 6c,d show a wide range of variability in methylation patterns between clones and subclones. What is the functional outcome of different methylation patterns?

9. The results section contains many parts which should ideally be placed in the methods section. The entire manuscript can be reduced to make it more concise.

10. Fig. 5c shows certain outliers from 90 kb onwards. It would be interesting to check the corresponding genes for these outliers. Also, the x axis does not extend until 200 kb, which is inconsistent with the text.

11. Related to the discussion section (page 15), there have been previous studies which have used COSMIC signature analysis in the context of induced pluripotent stem cells (such as D'Antonio et al. 2018 Cell Reports).

12. In general, the authors can state the p values also in the results section.

13. The authors should deposit their data on a public repository.

Reviewer #2:

Remarks to the Author:

Thompson and colleagues investigate the incidence of mutations and epigenetic stability in response to culture in the presence of a ROCK inhibitor (Y27632) and under low oxygen conditions in Shef11 hESC, compared to the incidence in Shef4 hESC maintained under standard conditions. They characterise a number of acquired mutations, with rates lower than those reported previously, although this may relate to their inclusion criteria. Significantly, they highlight that low oxygen culture is associated with a reduction in mutations, as well as with hypomethylation, suggesting culture conditions are likely important in the modulation of hESC genetic stability. As the stability of hESC over both short and long-term culture is a significant concern that impacts the potential of hESC for clinical application, their findings are timely and of importance.

Specific comments:

1. In their discussion, the authors note the importance of developing culture conditions that minimise changes in hESC lines. Given the low adoption of low oxygen culture worldwide, it would be useful to see more made of the significance of the reduced levels of mutations (and methylation) under these conditions.

2. My only concern with the study relates to establishing clones. The authors note that clones were screened and only those which were 'genetically normal' were carried forward. This would however establish an artificial horizon (of low mutation rates). The authors should provide a summary of the clones examined at each cloning point as to how many were deemed normal versus not and the types of mutations observed (did culture affect the incidence prior to selection?).

3. Related to this, the authors begin with lines at p36 and p15 (which are then passaged using manual cutting) for another 7 and 5 passages respectively before the first cloning event. Based on my calculations (from Fig 1 and a passaging frequency of ~ every 7 days as noted in the methods), approximately 16 passages were performed to expand clones under the different conditions (defining this as 'long-term culture'). Given that errors have been identified when cells have undergone more passages/at much later passage numbers than this, including the authors own previous reports, they should comment on whether the mutation rates reported are indeed

representative and indicative of the effects of long-term culture, or whether their snapshot in time is too early to detect more significant changes that may occur with further passage and expansion of these cells. Please also include the final passage numbers analysed for clarity (not just days in culture).

4. The authors highlight that the X chr showed an elevated mutation rate in all growth conditions. However, the authors have only analysed male lines. They should include discussion on the relevance of this finding with respect to both male and female lines.

5. It is of interest that while there is apparently less variability, the MShef4 clones display a higher number of mutations on average than the MShef11 clones (page 7). Given the authors statements around the stability of the lines used (Shef 4 being the more 'stable'), what do they make of this? Related to this, have the authors considered whether the differences at generation impact on stability (both variants of origin and acquired variants)? It is interesting to note that the more 'stable' Shef4 is derived from a frozen embryo, relative to a fresh embryo for Shef11. Freezing is known to damage cells, particularly within the embryo, and may disproportionately affect poorer quality cells, such that during isolation these less stable cells are lost, with only robust cells continuing.

It is also interesting to note that the two cell lines used were originally isolated under 5% conditions, but subsequently moved to 5%CO₂ in air for maintenance prior to experimentation. It would have been more useful to compare equivalently derived lines. As oxygen during derivation has been shown to modulate Xi (Lengner et al 2010), it would be interesting to speculate whether alternating O₂ conditions may have resulted in a level of destabilisation that promotes the acquisition of mutations and epigenetic change. Further, all clones from starting cultures were exposed to ROCKi to establish the parental clones. Perhaps these both contribute to the findings that all growth conditions showed some evidence of larger structural arrangements (particularly given that the authors later imply that the cloning process for Shef4 may have induced hypermethylation in the clones) and the decrease in methylation detected under low oxygen was less than that seen in the ICM and with naïve culture – i.e. handling effects?

6. It is a pity that experiments did not also investigate the effects of O₂ (or Y27632/ROCKi) on the 'more stable' MShef4 line (this may have led to more firm conclusions around methylation and low oxygen to have been possible). This is particularly valid given the authors highlight that one Shef11 clone (G8) was more responsive to oxygen than the other (I have presumed this refers to G2, based on Figure 1). This is not dissimilar to other reports which have suggested some hESC lines are not responsive to oxygen, which in itself could be significant. Is there any evidence from the parental clones that may indicate G2 arose from a less stable cell (this also relates to how many subclones were excluded from further analysis at each cloning point)? Confusingly, while Figure 1 suggests clone G2 was not expanded further, the authors note in the text (page 12) that both G2 and G8 were analysed following low O₂ culture and then +28 days 'normoxia' – both reversing their methylation profiles. Likewise, it is not clear in Figure 1 whether subclones N and Q were both examined +/- the 28 days of normoxia (although Figure 6a appears to suggest this).

7. The authors note that 5695 SNVs were detected in 4095 genes. However, no apparent 'genes of interest' were found. It is very difficult to distinguish the genes affected in the Suppl tables (3 and 4) provided – it would be useful to delineate these in some way for the reader – or provide those genes relating to specified mutations in the text (page 9 and 10) in a separate table. Was analysis of biological functions performed?

8. With regards to the reversion of methylation changes upon return to 20% conditions – what changes account for the 9% that did not reverse? When comparing WGBS and RNAseq data, beyond imprinted genes and ICRs, were other specific biological functions over-/under-represented?

9. As the authors note in their discussion that the decrease in methylation observed with low O₂ culture is less than that seen in cells extracted from the ICM they should include data demonstrating this comparison. It is otherwise unclear from the methods/results that these data have been compared with that available for the human blastocyst/inner cell mass. Do the profiles nevertheless suggest that the more hypomethylated signature observed in response to low O₂ is more ICM-like or do the changes maintained not equate to anything seen in the embryo? (This merely suggests that other additional factors are at play.)

Minor comments:

Page 6 'Mutation rate between cell lines and growth-conditions' results section: '(see Methods of details)' should read '(see Methods for details)'.

Page 9, delete the sentence 'Of these variants, one translocation involved chr 12....). The specifics of these are noted earlier in the same paragraph.

The authors note that they used flow to deposit single cells for clone mutation analysis. If surface markers were used during this process, they should be specified.

Figure 5d should be presented in the same, consistent order as used in other figures (Shef4, Shef11, Sheff11+Y27632, Shef11+5%O2).

Reference 36 is not the appropriate reference for the statement 'the effect of low oxygen on the behaviour and differentiation capacity of PSC has been studied extensively' as the few studies that exist have largely been published after 2008.

Reviewers' comments:

Reviewer #1 (Remarks to the Author):

Thompson and colleagues have studied the genetic and epigenetic changes occurring during in vitro culture of two clinical grade human embryonic stem cell (hESC) lines, and provided an important data to assess the safety of clinical application of hESC. The authors have generated a large number of datasets, and performed a thorough investigation to identify different types of mutations in hESC such as point mutations, insertions, deletions, and chromosomal translocations. The authors have discovered that the mutation rate was low in hESC during cell culture and that the mutation rate was reduced under low oxygen conditions. Epigenetic analysis has shown that clones grown in low oxygen conditions had elevated methylation compared to their parental controls in one cell line, although the functional consequence of this elevation is not shown. While this study might have limited scientific advancements, the authors have reported important findings to facilitate the translation of hESC to the clinic. However, the generality of the conclusions drawn from this study using only two hESC lines is uncertain. Listed below are some points of concern.

We are grateful to the reviewer for these comments recognising the extensive nature of our analyses and their potential significance in facilitating translation of hESC to the clinic. In this respect, we note that our observations on the underlying mutation rate in hESC, which is rather low, help to put into perspective the general concerns that are raised by our separate observations, and those of others, of the repetitive nature of the common genetic variants that occur during the culture and maintenance of hESC lines. Thus, hESC are not particularly genetically unstable but rather the commonly observed repetitive variants reflect the selective growth advantage of certain, most likely rare mutations. We understand the concern about the generality of our results and we address this and the reviewer's other specific comments below.

1. The authors have used only two hESC lines to draw their conclusions. The authors can mention in the text how commonly these cell lines are used. Importantly, how general is the conclusion drawn from this study (also considering the last paragraph of the discussion section on page 16, in which 41 lines out of 120 PSCs showed abnormal karyotype in the ISCI study)?

We appreciate the reviewer's concern about the generality of the conclusions that we have reached in the current study based upon just two hESC lines. Unfortunately for reasons of practicality and cost, as well as the depth of our study, which involved more than 80 WGS analyses, as well as WGBS and RNAseq, it was only feasible to analyse two hESC lines MShef4 and MShef11. Nevertheless, we note that the mutation rates, and other features of the observed mutations, in these two independent lines were not significantly different from one another, consistent with the view that the results are representative of hPSC in general. Although isolated in the same lab, MShef4 and MShef11 were derived from embryos obtained from different Assisted Conception Clinics, and so most likely of different parentage (because of confidentiality issues we are not allowed to know explicit details of the donors). To date these two lines have not been widely used in other studies (although they were included in the study of *TP53* mutations in hPSC by Merkle et al. 2017 *Nature* 545:229). However, they were derived to cGMP standards that are compatible with clinical use and they are generally available through the UK Stem Cell Bank.

The problem that we sought to address in our study, but is mostly not addressed in other published studies, is that by the time mutations become detectable it will have been necessary to expand a cell culture substantially, with the result that selection obscures the underlying mutation rate. In the ISCI study noted by the reviewer, the 41 lines that acquired abnormal karyotypes almost certainly reflects the selective growth advantages conferred by the karyotypic changes observed and provides no information about mutation rate. Nevertheless, that most of the lines (79) analysed in the ISCI project stayed karyotypically normal, while most of the more than 100 lines studied by Merkle et al 2017 (*Nature*

545:229), did not acquire TP53 mutations, can be regarded as consistent with our current estimates of a low underlying mutation rate in hPSC.

We are only aware of two studies that directly assess the rates, with respect to time, by which hPSC acquire mutations during culture, as opposed to assessing mutational burden, which necessarily depends on selection as well as mutation, or assessing pre-existing mutations and those induced by reprogramming. Using similar strategies to ours, these two studies estimated similar mutation rates to those that we have observed in MShef4 and MShef11: Rouhani et al 2016 (*PLoS Genet* 12, e1005932) estimated the mutation rates in two hiPSC and one hESC lines (H9) to be 0.18×10^{-9} SNV per bp per generation. In a more limited study of one hiPSC line, Kuijk et al. 2018 estimated a rate of 3.5 ± 0.5 bp substitutions per cell per population doubling – equivalent to about 1×10^{-9} SNV per bp per generation (<https://www.biorxiv.org/content/early/2018/09/29/430165.full.pdf>).

In both studies, similar to our results, the authors noted that C>A transversions predominated while Kuijk et al 2018 also noted that mutation signature 18 predominated, consistent with oxidative stress. Two other studies (*Kucab et al 2019 Cell* 177:821-836; *Zou et al 2018 Nat Commun.* 9:1744), though not addressing mutation rates, also reported that the background mutation pattern of C>A transversions is also similarly seen across many human iPSC clones. It is also notable that in an early study of a single locus, *Aprt*, the mutation rate in mouse ES cells was found to be substantially lower than in corresponding somatic cells (*Cervantes et al. 2002 PNAS* 99:3586).

Considering these points we have now modified Para 4 of the Introduction, and Para 1 and Para 3 of the Discussion, adding the additional references noted above, as well as additional information on the origins of the cell lines (as well as a link to the website of the UK Stem Cell Bank where the lines have been deposited) in the first section of the Materials and Methods, which we have also modified in response to the reviewer's point 9, below.

2. *The authors found that the mutation rate was higher in intergenic regions when compared to genic regions (Fig. 2c). The authors can perform deeper analysis in the context of epigenetic status, to examine how many mutations overlapped open chromatin and heterochromatin regions in hESC. The authors can overlay different histone marks and transcription factor binding to identify mutations present in active and repressed regions, similar to the analysis performed in Yoshihara et al. 2017 Cell Reports.*

We appreciate the reviewer's interest in mutation rates over a wider range of genomic features and regions of epigenetic sensitivity. To address this point we have expanded our analysis of mutation rates to examine more regions of interest; enhancers, transcription factor-binding sites, open chromatin, DNase I hypersensitive sites, CTCF repressor binding sites, CpG islands and 3' and 5' untranslated regions. We also included several histone modification marks in our analysis.

In all cases we observe a similar pattern of differences in mutation rates between groups, as observed in our original analysis of introns, exons, and intergenic regions. CpG islands appear to have a slightly elevated mutation rate compared to intergenic and intronic DNA, which is not surprising, given the susceptibility of these bases to mutation.

We have now included the expanded analysis in an additional Supplementary Figure 2b and added the following paragraph to the Results Section of the main text (Page 6):

"We also measured the mutation rates within a variety of other genomic features (Supplementary Figure 2b). In most cases, we found a similar pattern of difference between groups, with subclones derived from low oxygen parental clones displaying a significantly lower mutation rate than other groups. Two exceptions that showed slightly elevated mutation rates across all groups were CpG islands and transcription factor-binding sites (TFBS). Both showed a slightly higher mutation rate than other features (with the exception of intergenic regions), with no significant differences in mutation rates between groups,

suggesting that such genomic regions are particularly susceptible to mutation. However, given the small number of subclones that acquired mutations in these regions we could not draw further conclusions on the significance of this finding. The mutation rates over a variety of histone marks were also calculated (Supplementary Figure 2b). In the case of three types of histone mark (H3K4me1, H3K4me3, and H3K36me3), MShef4 had slightly higher though significant mutation rate than MShef11 grown under standard conditions ($P = 0.029$; $P = 0.008$; $P = 0.023$). In two cases (H3K9me3 and H3K27me3) MShef11 grown in low oxygen had a significantly lower mutation rate than MShef11 grown in standard conditions ($P = 0.0008$; $P = 0.013$)."

3. The authors reported a total of 225 SNVs in exons, including 7 nonsense mutations, and 85 (this number is not shown in the text) non-synonymous mutations. The authors have provided a complete list of mutations with their locations and other relevant details in the Supplementary Data 1. It would be nice if the authors can show the number of missense, nonsense mutations etc. in the form of a bar or pie chart. The authors can also provide more data about the mutated genes. What kind of genes are affected? This would help the readers better understand the data. Similarly, a figure can also be shown for number of mutations seen in 5' UTRs, coding exons, splice sites etc.

We agree with the reviewer's comment that it would be useful for the reader to be able to easily visualise the number and type of coding-region mutations detected in our analysis. Therefore, we have included a waffle chart showing these data in an additional **Supplementary Figure 2d**. Details of the genes and their positional information can be found in Supplementary Data 1 under individual tabs in the workbook. We also performed pathway analysis of each class of mutation but did not detect significant enrichment of any particular pathway or molecular function. (See also response to Point 4, below)

4. In Fig. 5b, it would be interesting to see which genes have a high mutational load with short gene lengths and low mutational load with long gene lengths. GO analysis might be also be helpful in this context. Also, the legend states "solid grey line" but I think the authors mean "solid black line".

We appreciate the reviewer's interest in the genes residing in the low and high mutational burden groups shown in Figure 5b. We have performed GO analysis on these groups of genes, using the PANTHER tool (Protein ANalysis THrough Evolutionary Relationships; (<http://www.pantherdb.org/>)). Briefly, we find enrichment of different pathways in both low-mutation (longer), and high-mutation (shorter) genes. However, given that these mutations occur across independent subclones and growth conditions, we cannot ascribe any particular significance to these enrichments. The results of this analysis is detailed in a new Supplementary Table 3. We have now added the following sentence to the manuscript (Page 9-10):

"GO analysis using the PANTHER (Protein ANalysis THrough Evolutionary Relationships; (<http://www.pantherdb.org/>)). Classification System highlighted some association of these genes with various signalling pathways, but as the mutations occurred across independent subclones and growth conditions, we cannot ascribe any particular significance to these enrichments (Supplementary Table 3) (Supplementary Data 5)."

We also thank the reviewer for pointing out the formatting error in Figure 5b, the description of which has now been corrected in the main text and figure legend.

5. The authors have generated many datasets using whole-genome sequencing (WGS), whole-genome bisulphite sequencing (WGBS) and RNA-seq. The authors have provided read statistics for WGBS and RNA-seq data, they can also provide the same for WGS data. Although the authors have integrated WGBS and RNA-seq data for imprint control regions analysis, this kind of

integrative analysis can be expanded. It would be nice if the authors can integrate WGS and RNA-seq data to study the functional effect of mutations on the transcriptome. For example, what effect do nonsense and splice site mutations have on the transcripts? Similarly, the authors can integrate the WGBS and RNA-seq data to understand the effect of differential methylation of CpG promoters (shown in Fig. 6b) on gene expression.

We apologise for omitting the WGS read statistics; also we note that we had not labelled the files for WGBS and RNAseq as 'Supplementary Data'. We have now provided the Read Statistics for the WGS, WGBS and RNAseq datasets into Supplementary Data 8, 9 and 10 respectively.

We appreciate the reviewer's interest in further integration of our sequencing datasets and the impact of coding site mutations on gene expression. To address, we have reviewed the expression levels of genes in those subclones with nonsense and splice region mutations and compared the level of expression to the remainder of the cohort with no such mutation. As a result, we have produced a new **Supplementary Table 2** comparing the expression levels of these mutated genes in the affected subclone with the expression levels in the non-mutated counterparts. We have also added the following paragraph to the main text (Page 8):

*"We further analysed whether nonsense mutations or mutations in splice sites had any effects on transcript levels (**Supplementary Table 2**). In only two of these mutations did we find evidence of reduced mRNA levels. MShel11 +Y27632 subclone K2 harbored a nonsense mutation in the gene DNAJC6, which exhibited a 43% decrease in expression compared to the mean of other subclones in this cohort (2.7 vs. 4.7), while MShel4 subclone J13 harbored a splice region mutation in the gene C11orf73, which exhibited a 32% decrease in expression compared to the mean of its cohort (8.4 vs. 12.34).*

We also appreciate the reviewer's comment in relation to integrating WGBS and RNAseq data. However, although we have analysed the correlation between gene expression and promoter methylation for both CpG island-containing and non-CpG island-containing promoters, in all cell lines and growth condition groups we found only a weak relationship between promoter methylation and gene expression, and this pattern was consistent across all groups. This 'weak' relationship between promoter methylation and control of gene expression is a general feature of PSC (Ficz, et al. 2013 Cell 13:351-359), which means that we could not draw any useful conclusions on the functional outcome of changes in gene expression in response to methylation status. We have added a sentence to make this point to the epigenetics section of the Results (Page 11):

"Overall we found a weak relationship between promoter methylation and gene expression."

For this reason we focused on imprinted genes and imprint control regions in Figure 6, which have a more robust functional relationship in hPSC.

6. The authors state that there was a positive correlation between gene expression and mutational load (Fig. 5d). What were the correlation coefficients and p values? The authors used log (FPKM) which is just natural log, it may be better to show gene expression values on a log2 scale.

We are grateful for the reviewer's comment requesting correlation coefficients and P-values relating to the relationship between mutational load and gene expression in each cell line/growth condition group. We have adjusted Figure 5d to include summary statistics on these correlations.

With regard to the reviewer's point on expressing gene expression on a log2 scale; we would prefer to maintain consistency in scaling across all figures in which gene expression

is measured. Since whichever scale is used does not affect the conclusion, we would like to maintain natural log as the scaling on the x-axis of Figure 5d.

7. The authors have mapped their sequencing data using two human genome assemblies (GRCh37 and 38). For consistency, it would be best to use one genome assembly.

We appreciate the reviewer's comment but, for practical reasons, we had to make some compromises in how we analysed our data. The various analyses performed in this manuscript involving a large number of whole genome analyses of the genetic, transcriptional and epigenetic makeup of the cells, would not have been feasible without the use of well-established computational pipelines and analysis tools. In particular, the genomic analyses are based on computational tools developed in previous projects and these tools had been implemented using the GRCh37 genome assembly. For our work to be comparable with previous studies and also to be able to fully use these pipelines, we were forced to map our genomic data to the GRCh37 genome build. Equally, we used a number of related tools to analyse the RNAseq data and so also used the GRCh37 assembly for those.

In contrast to this, the DNA methylation data was analysed using custom scripts and analysis which are not dependent on the genome build. Due to the technical challenges to map bisulphite data sets, any improvements in the genome build also result in improved mapping efficiencies and accuracy. Consequently, we decided to use the newest available genome build at the time of the analysis (which was GRCh38) and use that for all of our bisulphite analyses. Since many comparative analyses were done at the gene or promoter level, we could easily identify the relevant gene sets in the transcriptional data and match these to the relevant methylation data independently of the genome build.

To use one genome build for all analysis would have meant that we would have had to use the older genome build (GRCh37) for our methylation data, likely decreasing the mapping quality and as such lowering the quality of the analysis done. In our view, though for consistency it would seem neater to have used the same genome build throughout, in practice using different builds has not affected our conclusions.

8. Figs. 6c,d show a wide range of variability in methylation patterns between clones and subclones. What is the functional outcome of different methylation patterns?

We appreciate the reviewer's comment in relation to different patterns of methylation between clones and subclones. As mentioned in point 5, the relationship between methylation and gene expression is weak in hPSC, preventing an informative functional analysis of the changes observed between the parental clones and subclones. Further, due to the nature of the experimental design, we lack RNA-seq data from the parental clones, meaning that we were unable to attempt to correlate changes in both methylation and gene expression between clones growing under different conditions and the subsequent subclone cohorts.

9. The results section contains many parts which should ideally be placed in the methods section. The entire manuscript can be reduced to make it more concise.

We appreciate the Reviewer's comment and agree that certainly some elements that we had included in the Results Section would be better incorporated into the Materials and Methods. Accordingly, we have streamlined the first section on Experimental Strategy and the Legend to Figure 1, and incorporated these details into a revised section about the cell lines and cloning strategy in the Materials and Methods, Section "Cell Lines, culture and cloning methods".

10. Fig. 5c shows certain outliers from 90 kb onwards. It would be interesting to check the corresponding genes for these outliers. Also, the x axis does not extend until 200 kb, which is inconsistent with the text.

We are grateful to the reviewer for their point relating to the formatting of Figure 5c and their question relating to genes falling within outlier bins. To address this point, we have re-analysed our data using a refined computational approach from *Nik-Zainal et al. Cell 149, 979-993 (2012)*, which has both improved the precision of the analysis and has also allowed us to identify outliers in a statistically rigorous manner. This improved computational approach has altered the distribution of the data points slightly and revealed a positive correlation between increasing distance from TSS and the proportion of genes with mutation. This finding is very similar to that observed by Nik-Zainal and colleagues in their paper. We have edited Figure 5c accordingly and updated the text to reflect our analysis (Page 10):

"We found the prevalence of mutation increased with increasing distance from TSS (Figure 5c), as has been observed previously {Nik-Zainal, 2012 #25}."

We have also amended the scale of the x-axis in response to the reviewer's helpful observation.

11. Related to the discussion section (page 15), there have been previous studies which have used COSMIC signature analysis in the context of induced pluripotent stem cells (such as D'Antonio et al. 2018 Cell Reports).

We appreciate the point that others have also reported studies in which they have analysed mutational signatures in human PSC. All of these support the preponderance of C>A transversions, but, in our view, there is insufficient data to take the interpretations further. We have now added a sentence with references to D'Antonio et al 2018 and also Kucab et al 2019 to this paragraph (PAGE 14):

"In this study, cultures from all growth-conditions showed a strong correlation with a mutation signature frequently detected in cultured cells and linked to oxidative stress (COSMIC signature 18). Similar observations were also made in other studies of the mutational burden in human iPS cells (D'Antonio et al 2018; Kucab et al 2019)."

12. In general, the authors can state the p values also in the results section.

We have now stated p values in the text, in addition to figures and legends.

13. The authors should deposit their data on a public repository.

All of the sequencing data has been deposited in the public WTSC repository and EGS accession IDs have been provided for each data set. Details are provided in the Data accessibility Statement at the end of the Methods:

WGS: EGAS00001001561

Bisulphite Sequencing: EGAS00001001625

RNA-seq: EGAS00001001655

Reviewer #2 (Remarks to the Author):

Thompson and colleagues investigate the incidence of mutations and epigenetic stability in response to culture in the presence of a ROCK inhibitor (Y27632) and under low oxygen conditions in Shef11 hESC, compared to the incidence in Shef4 hESC maintained under standard conditions. They characterise a number of acquired mutations, with rates lower than those reported previously, although this may relate to their inclusion criteria. Significantly, they highlight that low oxygen

culture is associated with a reduction in mutations, as well as with hypomethylation, suggesting culture conditions are likely important in the modulation of hESC genetic stability. As the stability of hESC over both short and long-term culture is a significant concern that impacts the potential of hESC for clinical application, their findings are timely and of importance.

We appreciate the Reviewer's comments about the significance of our results and address the specific points raised, below.

Reviewer 2, Specific comments:

1. In their discussion, the authors note the importance of developing culture conditions that minimise changes in hESC lines. Given the low adoption of low oxygen culture worldwide, it would be useful to see more made of the significance of the reduced levels of mutations (and methylation) under these conditions.

We appreciate this point and have now added a sentence to the end of the second paragraph of the Discussion:

“Our results do suggest a benefit for maintaining human PSC cultures in a low oxygen environment, though to avoid exposure to fluctuating oxygen levels, for example to atmospheric oxygen during passaging and other manipulations, this requires the use of a dedicated low oxygen workstation, as we have done in this study.”

2. My only concern with the study relates to establishing clones. The authors note that clones were screened and only those which were ‘genetically normal’ were carried forward. This would however establish an artificial horizon (of low mutation rates). The authors should provide a summary of the clones examined at each cloning point as to how many were deemed normal versus not and the types of mutations observed (did culture affect the incidence prior to selection?).

We recognise the reviewer's concern. However, our intention for the early screening of the clones was not to carry out a detailed and comprehensive genetic review but rather just to eliminate any clones that might early on have acquired major structural changes in the genome, including those that are often seen in hPSC lines (because of their selective advantage) that might have compromised analysis of the WGS data. These initial screens were carried out within three passages of the cloning step, so the number of cells available was small: we used FISH to screen for common variants backed up with G-banding karyotyping SNP array analysis to detect other major structural changes. In the event, we detected very few genetic variants in these initial screens and to all intents and purposes the selection of which clones to take forward was effectively random. **In the Methods Section ‘Cell lines, culture and cloning methods’, we have now added a table, Supplementary Table 4, showing the results for the initial screening of clones.**

3. Related to this, the authors begin with lines at p36 and p15 (which are then passaged using manual cutting) for another 7 and 5 passages respectively before the first cloning event. Based on my calculations (from Fig 1 and a passaging frequency of ~ every 7 days as noted in the methods), approximately 16 passages were performed to expand clones under the different conditions (defining this as ‘long-term culture’). Given that errors have been identified when cells have undergone more passages/at much later passage numbers than this, including the authors own previous reports, they should comment on whether the mutation rates reported are indeed representative and indicative of the effects of long-term culture, or whether their snapshot in time is too early to detect more significant changes that may occur with further passage and expansion of these cells. Please also include the final passage numbers analysed for clarity (not just days in culture).

A key point of the strategy we followed in this study was to obtain estimates of the mutation rates in hESC, unbiased by selection. In most of the published studies, including our own previous studies, the analysis of mutations is confounded by the observation that those

mutations that are detected have only become detectable because they provide strong selective growth advantages and tend to overtake the wildtype cells - it is in that context that 'more significant changes may occur with further passage'. The presumption of our study is that the mutation rate is independent of time in culture and hence passage level; consequently, our strategy was to assess the mutations that occurred over a defined period, cultures starting from a single cell. The only obvious exception to this proposition would be if a mutant occurred that generated a mutator phenotype (e.g. by mutation in genes associated with DNA synthesis or repair). It seems that the frequency of such mutations would be inherently low and, indeed, we did not detect any mutations in genes that might lead to such a mutator phenotype. Nevertheless, as requested, we have included passage level as well as time in culture into the legend for Figure 1 and Methods, which have also been revised in accordance with Reviewer comment 1.9, above.

4. The authors highlight that the X chr showed an elevated mutation rate in all growth conditions. However, the authors have only analysed male lines. They should include discussion on the relevance of this finding with respect to both male and female lines.

We had already noted in the Discussion (para 4), that the higher mutation rate for the X chromosome might be due to the presence of just one X chromosome in the male lines. We have now modified our statement to emphasise that this might be specific to male lines:

"Indeed, we found that mutation was evenly distributed across the chromosomes, with the exception of the X chromosome, which exhibited a slightly elevated mutation rate compared to the genome-wide average. Because both MShef4 and MShef 11 are both male cell lines, it is possible that this is a male-specific phenomenon. For example, the absence of a second copy of the X chromosome for repair by homologous recombination, or increase transcription rates from X chromosomes in male PSC to compensate for the presence of two active X chromosomes in female PSC, may partly account for the elevated mutation rate."

5. It is of interest that while there is apparently less variability, the MShef4 clones display a higher number of mutations on average than the MShef11 clones (page 7). Given the authors statements around the stability of the lines used (Shef 4 being the more 'stable'), what do they make of this? Related to this, have the authors considered whether the differences at generation impact on stability (both variants of origin and acquired variants)? It is interesting to note that the more 'stable' Shef4 is derived from a frozen embryo, relative to a fresh embryo for Shef11. Freezing is known to damage cells, particularly within the embryo, and may disproportionately affect poorer quality cells, such that during isolation these less stable cells are lost, with only robust cells continuing.

We agree with the reviewer that these issues are of considerable interest. Of course, the methods by which PSC are derived or maintained may impact on the selective advantages of particular mutations and, for example, may impact on whether particular variants of origin show up in established lines. Freezing, for example, might impact on the selection of particular variants, but whether it affects the long term susceptibility of cells to mutation is moot. However, our focus in this study was rather to assess mutation rate in the absence of selection, something for which little direct data was available. As we discussed above, the scale of the study needed to tackle this particular question meant that it was not feasible to also carry out a more extensive comparison of different hPSC lines with the power needed to detect effects of derivation methods on mutation rate. Although the MShef4 line did show a higher number of mutations than MShef11, this was not statistically significant, which is consistent with the view that the method of derivation and genotype of the cells does not impact on the mutation rate of different lines, albeit with data from just two lines. Certainly, the issues raised by the reviewer are intriguing, but in our view we do not have enough data to make any meaningful comment, and they await future focused

studies on additional lines. We have now added some relevant comment to the first paragraph of the Discussion, also revised in part in response to Reviewer 1, Point 1,

5a. It is also interesting to note that the two cell lines used were originally isolated under 5% conditions, but subsequently moved to 5%CO₂ in air for maintenance prior to experimentation. It would have been more useful to compare equivalently derived lines. As oxygen during derivation has been shown to modulate Xi (Lengner et al 2010), it would be interesting to speculate whether alternating O₂ conditions may have resulted in a level of destabilisation that promotes the acquisition of mutations and epigenetic change. Further, all clones from starting cultures were exposed to ROCKi to establish the parental clones. Perhaps these both contribute to the findings that all growth conditions showed some evidence of larger structural arrangements (particularly given that the authors later imply that the cloning process for Shef4 may have induced hypermethylation in the clones) and the decrease in methylation detected under low oxygen was less than that seen in the ICM and with naïve culture – i.e. handling effects?

We are not sure exactly what the Reviewer means by these comments: He/she notes that the two lines were initially derived under low oxygen conditions (i.e. 'equivalent conditions'), and then suggests that "...it would have been more useful to compare equivalently derived lines." Certainly, although both lines were initially derived under 5% oxygen before transferring to norm-oxygen for routine maintenance once the cell lines were established, there were other differences in how the lines were derived – e.g. with and without feeders. However, as discussed above, as much as it would be interesting to know if cells derived in different ways do exhibit different sensitivities to mutation, it was not practicable to carry out the current design of experiment with a sufficiently large set of lines to make a meaningful analysis. In fact, the mutation rates for the two ES cell lines was not significantly different, and was similar to that reported by others for human iPSC lines as discussed above in response to Reviewer 1 Comment 1. It is also notable that in an early study of mutation rates in a single gene, *Aprt*, in the laboratory mouse, the authors also concluded that the mutation rate in ES cells was low and substantially less than in corresponding somatic cells. Accordingly, based on the available data, our current working hypothesis is that the method of derivation has little impact on susceptibility of PSC to mutation, and we have added further comments to this effect to the first paragraph of the Discussion. Of course further more focused studies might uncover more subtle effects, but that was beyond the scope of our current study.

The reviewer also points out that ROCKi was used in the initial isolation of the parent clones, and wonders whether this might have contributed to the appearance of a few large structural changes under all conditions. However, we only detected seven large rearrangements across all clones, far too few to make any meaningful statement about their cause.

6. It is a pity that experiments did not also investigate the effects of O₂ (or Y27632/ROCKi) on the 'more stable' MShef4 line (this may have led to more firm conclusions around methylation and low oxygen to have been possible). This is particularly valid given the authors highlight that one Shef11 clone (G8) was more responsive to oxygen than the other (I have presumed this refers to G2, based on Figure 1). This is not dissimilar to other reports which have suggested some hESC lines are not responsive to oxygen, which in itself could be significant. Is there any evidence from the parental clones that may indicate G2 arose from a less stable cell (this also relates to how many subclones were excluded from further analysis at each cloning point)? Confusingly, while Figure 1 suggests clone G2 was not expanded further, the authors note in the text (page 12) that both G2 and G8 were analysed following low O₂ culture and then +28 days 'normoxia' – both reversing their methylation profiles. Likewise, it is not clear in Figure 1 whether subclones N and Q were both examined +/- the 28 days of normoxia (although Figure 6a appears to suggest this).

Unfortunately, because of the scale, practical complexity and cost of the study we were only able to study two cell lines, and it wasn't feasible to analyse both cell lines under all

conditions. Consequently, we had to make a prior judgement of which conditions to test for each line. We now realise that in some aspects Figure 1 was not as clear as we intended: The issue with clone G2 was not that it did not grow, but rather that at the end of the three-month culture period, we failed to obtain any viable subclones. We think that this was due entirely to a chance technical problem, and we have no evidence from which to speculate that this in any way reflects G2 having been derived from a less stable cell. It was because of this and because we only successfully derived nine N subclones from the G8 clone, that we decided to derive a further set of subclones from G8, to ensure that we had a similar number of subclones from the low oxygen condition as for the other conditions for the mutation rate assessment. Unfortunately, by the time we realised that we needed to do this the G8 culture had been removed from the low oxygen condition, hence the +28 days in normoxia for the additional Q subclones. Because of the complexities and challenges of maintaining cells entirely in the low oxygen workstation, none of the N and Q subclones were expanded under low oxygen. **We have now revised Figures 1 and 6a to make these points more clearly.**

7. The authors note that 5695 SNVs were detected in 4095 genes. However, no apparent 'genes of interest' were found. It is very difficult to distinguish the genes affected in the Suppl tables (3 and 4) provided – it would be useful to delineate these in some way for the reader – or provide those genes relating to specified mutations in the text (page 9 and 10) in a separate table. Was analysis of biological functions performed?

We thank the reviewer for their comment on the layout of the gene mutation data. We would like to point out that the workbook 'Supplementary Data 1 – SNVs' contains individual tabs containing a breakdown of genes affected by various types of mutation. In these tabs we have listed the gene names as well as their positional information, base substitution, and more information on the effect on the transcript at RNA level and protein coding sequence (if relevant). This is also true for 'Supplementary Data 3 – INDELS'.

We also performed pathways analysis with statistical over-representation test of the genes associated to each mutation subcategory, using PANTHER (Protein ANalysis THrough Evolutionary Relationships) Classification System. No functional enrichment of exonic mutations was detected (Supplementary Data 1).

8. With regards to the reversion of methylation changes upon return to 20% conditions – what changes account for the 9% that did not reverse? When comparing WGBS and RNAseq data, beyond imprinted genes and ICRs, were other specific biological functions over-/under-represented?

We do not fully understand the reviewer's first point. However, we believe it relates to Figure 6a illustrating the reduction in global methylation levels of parental clones grown in 5% oxygen, compared to the subclones derived from that parental clone. As the analysis was performed by analysing global methylation of contiguous tiled regions across the genome, we cannot ascribe any particular mechanism or reason for changes that did not reverse. **However, we have now altered the labelling of Figure 6a in an attempt to improve the figure and clarify differences between groups shown in the plot.**

We also appreciate the reviewer's comment in relation to integrating WGBS and RNAseq data beyond analysis of imprinted genes and imprint control regions. In response, we have measured the correlation between gene expression and promoter methylation for both CpG island-containing and non-CpG island-containing promoters. In all cell lines and growth condition groups we find only a weak relationship between promoter methylation and gene expression, and this pattern is consistent across all groups. This 'weak' relationship between promoter methylation and control of gene expression is a feature of hPSC, which meant that we could not reliably draw any conclusions on the enrichment of genes involved in other biological functions, in which changes in methylation had occurred.

9. As the authors note in their discussion that the decrease in methylation observed with low O₂ culture is less than that seen in cells extracted from the ICM they should include data demonstrating this comparison. It is otherwise unclear from the methods/results that these data have been compared with that available for the human blastocyst/inner cell mass. Do the profiles nevertheless suggest that the more hypomethylated signature observed in response to low O₂ is more ICM-like or do the changes maintained not equate to anything seen in the embryo? (This merely suggests that other additional factors are at play.)

We thank the reviewer for the comment and we have now included a reference and comparison in the text regarding the differences between hESCs cultured in low O₂ and naïve hESCs or ICM cells. Briefly, the reduction of mean CpG methylation observed upon low O₂ culture in this study is 3.6% while naïve hESCs or ICM cells have 40-50% less CpG methylation than conventional hESCs. Given that in conventional hESC 80-90% of CpGs are methylated, the loss of 3.6% is only small. As such, we also found that the comparison of the regions changing in methylation between normal and low oxygen conditions with those regions being differentially methylated between naïve or ICM cells and conventional hESCs was not informative as basically the whole genome is hypomethylated in naïve hESCs or ICM cells. Based on this, we cannot verify whether low O₂ is a driver for low CpG methylation in the early embryo or whether other factors are contributing to this. Nonetheless, previous work from us and colleagues suggests that the DNA methylation machinery shows some key differences in the early embryo which is a key driver for the hypomethylated phenotype of naïve hESCs, primordial germ cells and ICM cells (e.g. von Meyenn et al 2016 Mol Cell 62:848-861).

We have modified Discussion Para 5 (Page 15) with the addition of the following sentence:

“However, the small decrease in methylation detected in this study (mean reduction of 3.6% in CpG methylation in low oxygen cultures) is much less than the decrease seen in cells extracted from the ICM or following naïve reprogramming (reduction of ~40% in CpG methylation in ICM or naïve human ES cells (Guo et al 2016)).”

Minor comments:

Page 6 ‘Mutation rate between cell lines and growth-conditions’ results section: ‘(see Methods of details)’ should read ‘(see Methods for details)’.

We have corrected this.

Page 9, delete the sentence ‘Of these variants, one translocation involved chr 12....). The specifics of these are noted earlier in the same paragraph.

We have deleted this sentence as requested.

The authors note that they used flow to deposit single cells for clone mutation analysis. If surface markers were used during this process, they should be specified.

We did not use cell surface markers during single cell deposition. It was not necessary for the purposes of this study as, in our experience in practice, only undifferentiated ES cells will form long term colonies and subsequent cultures after single cell deposition. This was confirmed by the transcriptome of the clones analysed, revealed by RNAseq.

Figure 5d should be presented in the same, consistent order as used in other figures (Shef4, Shef11, Shef11+Y27632, Shef11+5%O₂).

We have revised Figure 5d as requested

Reference 36 is not the appropriate reference for the statement 'the effect of low oxygen on the behaviour and differentiation capacity of PSC has been studied extensively' as the few studies that exist have largely been published after 2008.

We appreciate the reviewer's point. We have added references (*Fynes et al 2014 Stem Cells Dev. 23:1910-1922; Närvä et al 2013 PLoS One. 8:e78847*) and changed the text (Page 14) to read: "...although the effect of low oxygen on the behaviour and differentiation capacity of PSC has been reported (e.g. Fynes et al 2014; Närvä et al 2013)".

Reviewers' Comments:

Reviewer #1:

Remarks to the Author:

I thank the authors for kindly and adequately addressing my comments. Although the authors analyzed only two hESC clones due to a technical and financial reason, based on the modified Para 4 of the introduction, I see that the conclusion drawn in this study and other published studies suggest that hPSCs have lower mutations rates comparing to those of somatic cells. Although the previous study by Rouhani and colleagues might limit the novelty of this study, I understand the importance of the above conclusion in facilitating the translation of hESC to the clinic. The authors also report thorough integrative analysis of multi-layered omics data, and provide several interesting findings. It is important to note that the authors further showed that the mutation rate can be reduced by >50% under low oxygen conditions. The quality of Supplementary Figure 2 and 3 can still be improved.

Reviewer #2:

Remarks to the Author:

The authors have addressed the reviewer comments from the original review. In particular, it is appreciated that the considerable analyses comprising the study makes the analysis of additional lines, and other factors, quite prohibitory. It would have been useful to have included a more commonly used line in their analyses, though it would be statistically inappropriate to accommodate this post hoc.

The authors however rationalized that they were interested in two lines of differing quality (i.e. only 1 representative of each) yet subsequently find no difference. For me it remains a concern as to whether this lack of difference is real or merely relates to their bias in removing those clones displaying early genetic instability, thereby obtaining a falsely lower mutation rate.

Given their clone selection, I also question the relevance of the data to the wider stem cell community, where the vast majority of hESC lines in use have been heavily selected through bulk culture (i.e. display the growth advantage the authors have specifically excluded from their study). It is perhaps the clones that were excluded that may have provided more interesting and relevant information. Nonetheless, their data are of interest to the field, and pertinent for their application clinically.

Oliver Thompson et al, “Low rates mutations in clinical grade human pluripotent stem cells under different culture conditions”

Response to Final Comments of Reviewers

Reviewer 1: *I thank the authors for kindly and adequately addressing my comments. Although the authors analyzed only two hESC clones due to a technical and financial reason, based on the modified Para 4 of the introduction, I see that the conclusion drawn in this study and other published studies suggest that hPSCs have lower mutations rates comparing to those of somatic cells. Although the previous study by Rouhani and colleagues might limit the novelty of this study, I understand the importance of the above conclusion in facilitating the translation of hESC to the clinic. The authors also report thorough integrative analysis of multi-layered omics data, and provide several interesting findings. It is important to note that the authors further showed that the mutation rate can be reduced by >50% under low oxygen conditions. The quality of Supplementary Figure 2 and 3 can still be improved*

We appreciate the reviewer's final comments. In the final submitted manuscript we have improved the quality of Supplementary Figures 2 and 3 (now Supplementary Figures 5 and 6, following renumbering in response to the Editor's request.)

Reviewer 2: *The authors have addressed the reviewer comments from the original review. In particular, it is appreciated that the considerable analyses comprising the study makes the analysis of additional lines, and other factors, quite prohibitory. It would have been useful to have included a more commonly used line in their analyses, though it would be statistically inappropriate to accommodate this post hoc.*

The authors however rationalized that they were interested in two lines of differing quality (i.e. only 1 representative of each) yet subsequently find no difference. For me it remains a concern as to whether this lack of difference is real or merely relates to their bias in removing those clones displaying early genetic instability, thereby obtaining a falsely lower mutation rate.

Given their clone selection, I also question the relevance of the data to the wider stem cell community, where the vast majority of hESC lines in use have been heavily selected through bulk culture (i.e. display the growth advantage the authors have specifically excluded from their study). It is perhaps the clones they were excluded that may have provided more interesting and relevant information. Nonetheless, their data are of interest to the field, and pertinent for their application clinically.

We appreciate the reviewer's comments. Unfortunately, the clonal nature of the experimental design, which we adopted to try to obviate the problem of selective growth advantages of individual mutants confounding estimates of the underlying mutation rate, made the practicalities of carrying out the experiments particularly challenging, so that we had to accept various compromises to make the experiments practicable. We understand the point about selecting the clones for study. However, one of our initial concerns at the beginning of the experiment was that early acquisition of gross genomic changes in a parent clone might compromise later analysis of whole genome data. Consequently, as we could only practically analyse the subclones of one or two parental clones under each condition, we opted to carry out the initial screen and selection we described. To address these issues, taking advice as suggested by the editor, we have made changes as follows:

1. We have rewritten the abstract, which now reads:

The occurrence of repetitive genomic changes that provide a selective growth advantage in pluripotent stem cells is of concern for their clinical application. However, the effect of different culture conditions on the underlying mutation rate is unknown. Here we show that the mutation rate in two human embryonic stem cell lines derived and banked for clinical application is low and not significantly affected by culture with Rho Kinase inhibitor, commonly used in their routine maintenance.

However, the mutation rate is reduced by >50% in cells cultured under 5% oxygen, when we also found alterations in imprint methylation and reversible DNA hypomethylation. Mutations are evenly distributed across the chromosomes, except for a slight increase on the X-chromosome, and an elevation in intergenic regions suggesting that chromatin structure may affect mutation rate. Overall the results suggest that pluripotent stem cells are not subject to unusually high rates of genetic or epigenetic alterations.

2. We have included the following, highlighted in yellow, into the first paragraph of the Discussion:

Understanding genetic and epigenetic change in human PSC during in vitro culture is important in the context of regenerative medicine, as these cells represent a starting point for stem cell-based therapeutics. Thus, developing culture conditions that minimise change is important for improving safety in clinical applications of human PSC-derived therapeutics. It is encouraging that following culture over an extended period of time and multiple passages, both the human PSC in this study acquired a relatively low burden of single-nucleotide base substitutions, and none of the common genetic variation seen in human PSC. This is consistent with the view that the common non-random genetic changes observed in human PSC are driven by the selective growth advantage of rare but random mutations. However, estimating the underlying rate of mutation in PSC is difficult because by the time mutants become detectable their frequency may be grossly distorted by the effects of selection of those mutants that offer a growth advantage. The clonogenic strategy that we adopted to obviate this difficulty is, however, complex and expensive, so that it was only feasible to analyse two human ES cell lines, and the necessary production and selection of clones with which to carry out the analysis might itself have introduced distortions. Nevertheless, the rates we observed under normal growth conditions (0.37×10^{-9} and 0.28×10^{-9} SNV per day, for genetically independent human ES cell lines (MShef4 and MShef11, respectively) are not significantly different and are comparable with two other recent studies of human PSC (0.18×10^{-9} and 1×10^{-9} SNV per base-pair, per cell division, respectively)³¹⁸. They substantially lower than the mutation rates estimated in somatic cells. The two ES cell lines used in our present study are mostly likely genetically unrelated, and one was derived from a frozen embryo and one from a fresh embryo. By contrast the lines studied by Rouhani were iPS cell lines, derived by reprogramming. That the mutation rate in all of these lines was comparably low, suggests that this may be a feature of PSC in general, irrespective of their means of derivation. It is also notable that in a study of a single locus, Aprt, in mouse ES cells,³³ concluded that the mutation rate in mouse ES cells is substantially lower than in corresponding somatic cells.

3. In the final paragraph of the discussion we have added a further sentence, highlighted in yellow:

Overall, the striking conclusion from this study is the low mutation rate in human PSC, whether affecting SNV or INDELS, despite the frequent reports of common genetic variants in the literature. Most likely, the latter reflects an ascertainment bias. In the ISCI study of 120 pairs of human PSC in early and late passage, 79 lines remained karyotypically normal while in a sequencing study of 140 human ES cell lines 12, only six acquired mutations in TP53, all results consistent with an underlying low mutation rate in human PSC. Of course, one unknown is whether PSC lines that have acquired growth advantages through long periods in culture may have an altered mutation rate, perhaps a mutator phenotype. Evidently, the mutation burden in human PSC can be reduced by culture conditions, such in a low oxygen environment, but it seems that the appearance of common variants is largely a consequence of selection rather than underlying mutation. Minimising the appearance of such variants will, then, depend primarily upon identification and moderating of the mechanisms by which they exhibit a growth advantage.